# Accelerated Policy Learning with Parallel Differentiable Simulation

**Jie Xu**[1,2]**, Viktor Makoviychuk**[1]**, Yashraj Narang**[1]**, Fabio Ramos**[1,3]**,
Wojciech Matusik**[2]**, Animesh Garg**[1,4]**, Miles Macklin**[1]

[1]NVIDIA  [2]Massachusetts Institute of Technology  [3]University of Sydney  [4]University of Toronto

## Abstract

Deep reinforcement learning can generate complex control policies, but requires large amounts of training data to work effectively. Recent work has attempted to address this issue by leveraging differentiable simulators. However, inherent problems such as local minima and exploding/vanishing numerical gradients prevent these methods from being generally applied to control tasks with complex contact-rich dynamics, such as humanoid locomotion in classical RL benchmarks. In this work we present a high-performance differentiable simulator and a new policy learning algorithm (SHAC) that can effectively leverage simulation gradients, even in the presence of non-smoothness. Our learning algorithm alleviates problems with local minima through a smooth critic function, avoids vanishing/exploding gradients through a truncated learning window, and allows many physical environments to be run in parallel. We evaluate our method on classical RL control tasks, and show substantial improvements in sample efficiency and wall-clock time over state-of-the-art RL and differentiable simulation-based algorithms. In addition, we demonstrate the scalability of our method by applying it to the challenging high-dimensional problem of muscle-actuated locomotion with a large action space, achieving a greater than $17\times$ reduction in training time over the best-performing established RL algorithm. More visual results are provided at: https://short-horizon-actor-critic.github.io/.

## 1 Introduction

Learning control policies is an important task in robotics and computer animation. Among various policy learning techniques, reinforcement learning (RL) has been a particularly successful tool to learn policies for systems ranging from robots (*e.g.,* Cheetah, Shadow Hand) (Hwangbo et al., 2019; Andrychowicz et al., 2020) to complex animation characters (*e.g.,* muscle-actuated humanoids) (Lee et al., 2019) using only high-level reward definitions. Despite this success, RL requires large amounts of training data to approximate the policy gradient, making learning expensive and time-consuming, especially for high-dimensional problems (Figure 1, Right). The recent development of differentiable simulators opens up new possibilities for accelerating the learning and optimization of control policies. A differentiable simulator may provide accurate first-order gradients of the task performance reward with respect to the control inputs. Such additional information potentially allows the use of efficient gradient-based methods to optimize policies. As recently Freeman et al. (2021) show, however, despite the availability of differentiable simulators, it has not yet been convincingly demonstrated that they can effectively accelerate policy learning in complex high-dimensional and contact-rich tasks, such as some traditional RL benchmarks. There are several reasons for this:

1. Local minima may cause gradient-based optimization methods to stall.
2. Numerical gradients may vanish/explode along the backward path for long trajectories.
3. Discontinuous optimization landscapes can occur during policy failures/early termination.

Because of these challenges, previous work has been limited to the optimization of open-loop control policies with short task horizons (Hu et al., 2019; Huang et al., 2021), or the optimization of policies for relatively simple tasks (*e.g.,* contact-free environments) (Mora et al., 2021; Du et al., 2021). In this work, we explore the question: *Can differentiable simulation accelerate policy learning in tasks with continuous closed-loop control and complex contact-rich dynamics?*

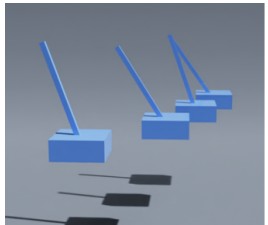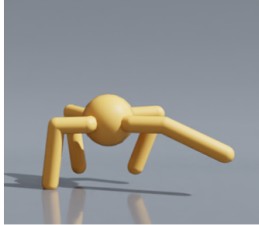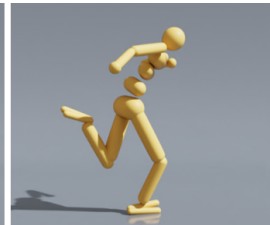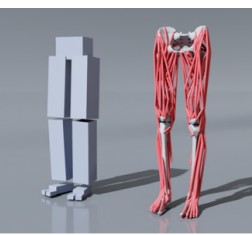

Figure 1: **Environments**: Here are some of our environments for evaluation. Three classical physical control RL benchmarks of increasing difficulty, from left: Cartpole Swing Up + Balance, Ant, and Humanoid. In addition, we train the policy for the high-dimensional muscle-tendon driven Humanoid MTU model from Lee et al. (2019). Whereas model-free reinforcement learning (PPO, SAC) needs many samples for such high-dimensional control problems, SHAC scales efficiently through the use of analytic gradients from differentiable simulation with a parallelized implementation, both in sample complexity and wall-clock time.

Inspired by actor-critic RL algorithms (Konda & Tsitsiklis, 2000), we propose an approach to effectively leverage differentiable simulators for policy learning. We alleviate the problem of local minima by using a critic network that acts as a smooth surrogate to approximate the underlying noisy reward landscape resulted by complex dynamics and occurrences of policy failures (Figure 2). In addition, we use a truncated learning window to shorten the backpropagation path to address problems with vanishing/exploding gradients, reduce memory requirements, and improve learning efficiency.

A further challenge with differentiable simulators is that the backward pass typically introduces some computational overhead compared to optimized forward-dynamics physics engines. To ensure meaningful comparisons, we must ensure that our learning method not only improves sample-efficiency, but also wall-clock time. GPU-based physics simulation has shown remarkable effectiveness for accelerating model-free RL algorithms (Liang et al., 2018; Allshire et al., 2021), given this, we develop a GPU-based *differentiable* simulator that can compute gradients of standard robotics models over many environments in parallel. Our PyTorch-based simulator allows us to connect high-quality simulation with existing algorithms and tools.

To the best of our knowledge, this work is the first to provide a fair and comprehensive comparison between gradient-based and RL-based policy learning methods, where fairness is defined as (a) benchmarking on both RL-favored tasks and differentiable-simulation-favored tasks, (b) testing complex tasks (*i.e.*, contact-rich tasks with long task horizons), (c) comparing to the state-of-the-art implementation of RL algorithms, and (d) comparing both sample efficiency and wall-clock time. We evaluate our method on standard RL benchmark tasks, as well as a high-dimensional character control task with over 150 actuated degrees of freedom (some of tasks are shown in Figure 1). We refer to our method as Short-Horizon Actor-Critic (SHAC), and our experiments show that SHAC outperforms state-of-the-art policy learning methods in both sample-efficiency and wall-clock time.

## 2 RELATED WORK

**Differentiable Simulation** Physics-based simulation has been widely used in the robotics field (Todorov et al., 2012; Coumans & Bai, 2016). More recently, there has been interest in the construction of differentiable simulators, which directly compute the gradients of simulation outputs with respect to actions and initial conditions. These simulators may be based on auto-differentiation frameworks (Griewank & Walther, 2003; Heiden et al., 2021; Freeman et al., 2021) or analytic gradient calculation (Carpentier & Mansard, 2018; Geilinger et al., 2020; Werling et al., 2021). One challenge for differentiable simulation is the non-smoothness of contact dynamics, leading many works to focus on how to efficiently differentiate through linear complementarity (LCP) models of contact (Degrave et al., 2019; de Avila Belbute-Peres et al., 2018; Werling et al., 2021) or leverage a smooth penalty-based contact formulation (Geilinger et al., 2020; Xu et al., 2021).

**Deep Reinforcement Learning** Deep reinforcement learning has become a prevalent tool for learning control policies for systems ranging from robots (Hwangbo et al., 2019; OpenAI et al., 2019; Xu et al., 2019; Lee et al., 2020; Andrychowicz et al., 2020; Chen et al., 2021a), to complex animation characters (Peng et al., 2018; 2021; Liu & Hodgins, 2018; Lee et al., 2019). Model-free RL algorithms treat the underlying dynamics as a black box in the policy learning process. Among them, on-policy RL approaches (Schulman et al., 2015; 2017) improve the policy from the experience generated by the current policy, while off-policy methods (Lillicrap et al., 2016; Mnih et al.,

2016; Fujimoto et al., 2018; Haarnoja et al., 2018) leverage all the past experience as a learning resource to improve sample efficiency. On the other side, model-based RL methods (Kurutach et al., 2018; Janner et al., 2019) have been proposed to learn an approximated dynamics model from little experience and then fully exploit the learned dynamics model during policy learning. The underlying idea of our method shares similarity with some prior model-based methods (Hafner et al., 2019; Clavera et al., 2020) that both learn in an actor-critic mode and leverage the gradient of the simulator "model". However, we show that our method can achieve considerably better wall-clock time efficiency and policy performance using parallel differentiable simulation in place of a learned model. A detailed note on comparison with model-based RL is presented in Appendix A.4.6.

**Differentiable Simulation based Policy Learning** The recent development of differentiable simulators enables the optimization of control policies via the provided gradient information. Backpropagation Through Time (BPTT) (Mozer, 1995) has been widely used in previous work to showcase differentiable systems (Hu et al., 2019; 2020; Liang et al., 2019; Huang et al., 2021; Du et al., 2021). However, the noisy optimization landscape and exploding/vanishing gradients in long-horizon tasks make such straightforward first-order methods ineffective. A few methods have been proposed to resolve this issue. Qiao et al. (2021) present a sample enhancement method to increase RL sample-efficiency for the simple MuJoCo Ant environment. However, as the method follows a model-based learning framework, it is significantly slower than state-of-the-art on-policy methods such as PPO (Schulman et al., 2017). Mora et al. (2021) propose interleaving a trajectory optimization stage and an imitation learning stage to detach the policy from the computation graph in order to alleviate the exploding gradient problem. They demonstrate their methods on simple control tasks (*e.g.,* stopping a pendulum). However, gradients flowing back through long trajectories of states can still create challenging optimization landscapes for more complex tasks. Furthermore, both methods require the full simulation Jacobian, which is not commonly or efficiently available in reverse-mode differentiable simulators. In contrast, our method relies only on first-order gradients. Therefore, it can naturally leverage simulators and frameworks that can provide this information.

## 3 METHOD

### 3.1 GPU-BASED DIFFERENTIABLE DYNAMICS SIMULATION

Conceptually, we treat the simulator as an abstract function $\mathbf{s}_{t+1} = \mathcal{F}(\mathbf{s}_t, \mathbf{a}_t)$ that takes a state $\mathbf{s}$ from a time $t \rightarrow t+1$, where $\mathbf{a}$ is a vector of actuation controls applied during that time-step (may represent joint torques, or muscle contraction signals depending on the problem). Given a differentiable scalar loss function $\mathcal{L}$, and its adjoint $\mathcal{L}^* = \frac{\partial \mathcal{L}}{\partial \mathbf{s}_{t+1}}$, the simulator backward pass computes:

$$\frac{\partial \mathcal{L}}{\partial \mathbf{s}_t} = \left(\frac{\partial \mathcal{L}}{\partial \mathbf{s}_{t+1}}\right)\left(\frac{\partial \mathcal{F}}{\partial \mathbf{s}_t}\right), \quad \frac{\partial \mathcal{L}}{\partial \mathbf{a}_t} = \left(\frac{\partial \mathcal{L}}{\partial \mathbf{s}_{t+1}}\right)\left(\frac{\partial \mathcal{F}}{\partial \mathbf{a}_t}\right) \tag{1}$$

Concatenating these steps allows us to propagate gradients through an entire trajectory.

To model the dynamics function $\mathcal{F}$, our physics simulator solves the forward dynamics equations

$$\mathbf{M}\ddot{\mathbf{q}} = \mathbf{J}^T\mathbf{f}(\mathbf{q}, \dot{\mathbf{q}}) + \mathbf{c}(\mathbf{q}, \dot{\mathbf{q}}) + \boldsymbol{\tau}(\mathbf{q}, \dot{\mathbf{q}}, \mathbf{a}), \tag{2}$$

where $\mathbf{q}, \dot{\mathbf{q}}, \ddot{\mathbf{q}}$ are joint coordinates, velocities and accelerations respectively, $\mathbf{f}$ represents external forces, $\mathbf{c}$ includes Coriolis forces, and $\boldsymbol{\tau}$ represents joint-space actuation. The mass matrix $\mathbf{M}$, and Jacobian $\mathbf{J}$, are computed in parallel using one thread per-environment. We use the composite rigid body algorithm (CRBA) to compute articulation dynamics which allows us to cache the resulting matrix factorization at each step (obtained using parallel Cholesky decomposition) for re-use in the backwards pass. After determining joint accelerations $\ddot{\mathbf{q}}$ we perform a semi-implicit Euler integration step to obtain the updated system state $\mathbf{s} = (\mathbf{q}, \dot{\mathbf{q}})$.

For simple environments we actuate our agents using torque-based control, in which the policy outputs $\boldsymbol{\tau}$ at each time-step. For the more complex case of muscle-actuation, each muscle consists of a list of attachment sites that may pass through multiple rigid bodies, and the policy outputs activation values for each muscle. A muscle activation signal generates purely contractive forces (mimicking biological muscle) with maximum force prescribed in advance (Lee et al., 2019).

Analytic articulated dynamics simulation can be non-smooth and even discontinuous when contact and joint limits are introduced, and special care must be taken to ensure smooth dynamics. To model contact, we use the frictional contact model from Geilinger et al. (2020), which approximates Coulomb friction with a linear step function, and incorporate the contact damping force formulation

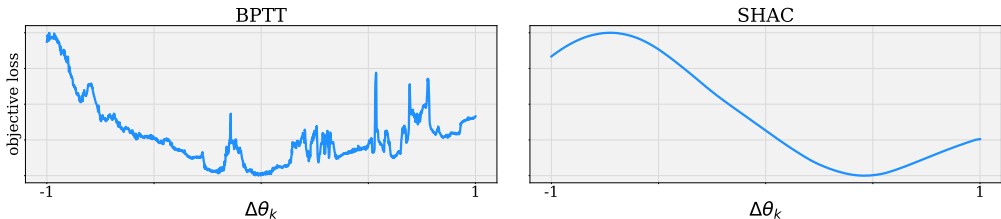

Figure 2: **Landscape comparison between BPTT and SHAC.** We select one single weight from a policy and change its value by $\Delta\theta_k \in [-1, 1]$ to plot the task loss landscapes of BPTT and SHAC w.r.t. one policy parameter. The task horizon is $H = 1000$ for BPTT, and the short horizon length for our method is $h = 32$. As we can see, longer optimization horizons lead to noisy loss landscape that are difficult to optimize, and the landscape of our method can be regarded as a smooth approximation of the real landscape.

from Xu et al. (2021) to provide better smoothness of the non-interpenetration contact dynamics:

$$\mathbf{f}_c = (-k_n + k_d \dot{d}) \min(d, 0)\mathbf{n}, \quad \mathbf{f}_t = -\frac{\mathbf{v}_t}{\|\mathbf{v}_t\|} \min(k_t\|\mathbf{v}_t\|, \mu\|\mathbf{f}_c\|) \tag{3}$$

where $\mathbf{f}_c, \mathbf{f}_t$ are the contact normal force and contact friction force respectively, $d$ and $\dot{d}$ are the contact penetration depth and its time derivative (negative for penetration), $\mathbf{n}$ is the contact normal direction, $\mathbf{v}_t$ is the relative speed at the contact point along the contact tangential direction, and $k_n, k_d, k_t, \mu$ are contact stiffness, contact damping coefficient, friction stiffness and coefficient of friction respectively.

To model joint limits, a continuous penalty-based force is applied:

$$\mathbf{f}_{\text{limit}} = \begin{cases} k_{\text{limit}}(q_{\text{lower}} - q), & q < q_{\text{lower}} \\ k_{\text{limit}}(q_{\text{upper}} - q), & q > q_{\text{upper}} \end{cases} \tag{4}$$

where $k_{\text{limit}}$ is the joint limit stiffness, and $[q_{\text{lower}}, q_{\text{upper}}]$ is the bound for the joint angle $q$.

We build our differentiable simulator on PyTorch (Paszke et al., 2019) and use a source-code transformation approach to generate forward and backward versions of our simulation kernels (Griewank & Walther, 2003; Hu et al., 2020). We parallelize the simulator over environments using distributed GPU kernels for the dense matrix routines and evaluation of contact and joint forces.

### 3.2 OPTIMIZATION LANDSCAPE ANALYSIS

Although smoothed physical models improve the local optimization landscape, the combination of forward dynamics and the neural network control policy renders each simulation step non-linear and non-convex. This problem is exacerbated when thousands of simulation steps are concatenated and the actions in each step are coupled by a feedback control policy. The complexity of the resulting reward landscape leads simple gradient-based methods to easily become trapped in local optima.

Furthermore, to handle agent failure (*e.g.,* a humanoid falling down) and improve sample efficiency, early termination techniques are widely used in policy learning algorithms (Brockman et al., 2016). Although these have proven effective for model-free algorithms, early termination introduces additional discontinuities to the optimization problem, which makes methods based on analytical gradients less successful.

To analyze this problem, inspired by previous work (Parmas et al., 2018), we plot the optimization landscape in Figure 2 (Left) for a humanoid locomotion problem with a 1000-step task horizon. Specifically, we take a trained policy, perturb the value of a single parameter $\theta_k$ in the neural network, and evaluate performance for the policy variations. As shown in the figure, with long task horizons and early termination, the landscape of the humanoid problem is highly non-convex and discontinuous. In addition, the norm of the gradient $\frac{\partial \mathcal{L}}{\partial \theta}$ computed from backpropagation is larger than $10^6$. We provide more landscape analysis in Appendix A.5. Thus, most previous works based on differentiable simulation focus on short-horizon tasks with contact-free dynamics and no early termination, where pure gradient-based optimization (*e.g.*, BPTT) can work successfully.

### 3.3 SHORT-HORIZON ACTOR-CRITIC (SHAC)

To resolve the aforementioned issues of gradient-based policy learning, we propose the Short-Horizon Actor-Critic method (SHAC). Our method concurrently learns a policy network (*i.e.*, actor) $\pi_\theta$ and a value network (*i.e.*, critic) $V_\phi$ during task execution, and splits the entire task horizon into

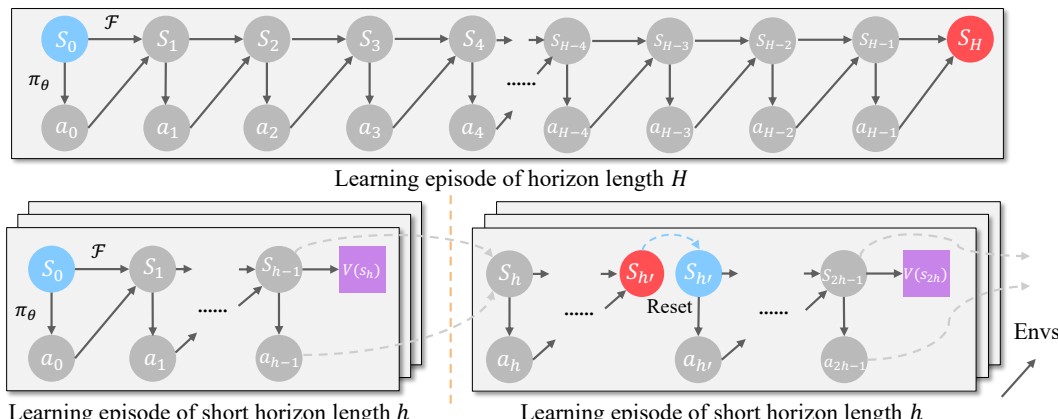

Figure 3: **Computation graph of BPTT and SHAC**. **Top**: BPTT propagates gradients through an entire trajectory in each learning episode. This leads to noisy loss landscapes, increased memory, and numerical gradient problems. **Bottom**: SHAC subdivides the trajectory into short optimization windows across learning episodes. This results in a smoother surrogate reward function and reduces memory requirements, enabling parallel sampling of many trajectories. The environment is reset upon early termination happens. Solid arrows denote gradient-preserving computations; dashed arrows denote locations at which the gradients are cut off.

several sub-windows of smaller horizons across learning episodes (Figure 3). A multi-step reward in the sub-window plus a terminal value estimation from the learned critic is used to improve the policy network. The differentiable simulation is used to backpropagate the gradient through the states and actions inside the sub-windows to provide an accurate policy gradient. The trajectory rollouts are then collected and used to learn the critic network in each learning episode.

Specifically, we model each of our control problems as a finite-horizon Markov decision process (MDP) with state space $\mathcal{S}$, action space $\mathcal{A}$, reward function $\mathcal{R}$, transition function $\mathcal{F}$, initial state distribution $\mathcal{D}_{\mathbf{s}_0}$, and task horizon $H$. At each step, an action vector $\mathbf{a}_t$ is computed by a feedback policy $\pi_\theta(\mathbf{a}_t|\mathbf{s}_t)$. While our method does not constrain the policy to be deterministic or stochastic, we use the stochastic policy in our experiments to facilitate extra exploration. Specifically, the action is sampled by $\mathbf{a}_t \sim \mathcal{N}(\mu_\theta(\mathbf{s}_t), \sigma_\theta(\mathbf{s}_t))$. The transition function $\mathcal{F}$ is modeled by our differentiable simulation (Section 3.1). A single-step reward $r_t = \mathcal{R}(\mathbf{s}_t, \mathbf{a}_t)$ is received at each step. The goal of the problem is to find the policy parameters $\theta$ that maximize the expected finite-horizon reward.

Our method works in an on-policy mode as follows. In each learning episode, the algorithm samples $N$ trajectories $\{\tau_i\}$ of short horizon $h \ll H$ in parallel from the simulation, which continue from the end states of the trajectories in the previous learning episode. The following policy loss is then computed:

$$\mathcal{L}_\theta = -\frac{1}{Nh} \sum_{i=1}^N \left[ \left( \sum_{t=t_0}^{t_0+h-1} \gamma^{t-t_0} \mathcal{R}(\mathbf{s}_t^i, \mathbf{a}_t^i) \right) + \gamma^h V_\phi(\mathbf{s}_{t_0+h}^i) \right], \tag{5}$$

where $\mathbf{s}_t^i$ and $\mathbf{a}_t^i$ are the state and action at step $t$ of the $i$-th trajectory, and $\gamma < 1$ is a discount factor introduced to stabilize the training. Special handling such as resetting the discount ratio is conducted when task termination happens during trajectory sampling.

To compute the gradient of the policy loss $\frac{\partial \mathcal{L}_\theta}{\partial \theta}$, we treat the simulator as a differentiable layer (with backward pass shown in Eq. 1) in the PyTorch computation graph and perform regular backpropagation. We apply reparameterization sampling method to compute the gradient for the stochastic policy. For details of gradient computation, see Appendix A.1. Our algorithm then updates the policy using one step of Adam (Kingma & Ba, 2014). The differentiable simulator plays a critical role here, as it allows us to fully utilize the underlying dynamics linking states and actions, as well as optimize the policy, producing better short-horizon reward inside the trajectory and a more promising terminal state for the sake of long-term performance. We note that the gradients are cut off between learning episodes to prevent unstable gradients during long-horizon backpropagation.

After we update the policy $\pi_\theta$, we use the trajectories collected in the current learning episode to train the value function $V_\phi$. The value function network is trained by the following MSE loss:

$$\mathcal{L}_\phi = \mathbb{E}_{\mathbf{s} \in \{\tau_i\}} \left[ \| V_\phi(\mathbf{s}) - \tilde{V}(\mathbf{s}) \|^2 \right], \tag{6}$$

---

**Algorithm 1:** SHAC (Short-Horizon Actor-Critic) Policy Learning

---

Initialize policy $\pi_\theta$, value function $V_\phi$, and target value function $V_{\phi'} \leftarrow V_\phi$.

**for** $learning\ episode \leftarrow 1, 2, ..., M$ **do**

    Sample $N$ short-horizon trajectories of length $h$ by the parallel differentiable simulation from the end states of the previous trajectories.

    Compute the policy loss $\mathcal{L}_\theta$ defined in Eq. 5 from the sampled trajectories and $V_{\phi'}$.

    Compute the analytical gradient $\frac{\partial \mathcal{L}_\theta}{\partial \theta}$ and update the policy $\pi_\theta$ one step with Adam.

    Compute estimated values for all the states in sampled trajectories with Eq. 7.

    Fit the value function $V_\phi$ using the critic loss defined in Eq. 6.

    Update target value function: $V_{\phi'} \leftarrow \alpha V_{\phi'} + (1 - \alpha) V_\phi$.

**end for**

---

where $\tilde{V}(\mathbf{s})$ is the estimated value of state $\mathbf{s}$, and is computed from the sampled short-horizon trajectories through a td-$\lambda$ formulation (Sutton et al., 1998), which computes the estimated value by exponentially averaging different $k$-step returns to balance the variance and bias of the estimation:

$$\tilde{V}(\mathbf{s}_t) = (1 - \lambda)\left( \sum_{k=1}^{h-t-1} \lambda^{k-1} G_t^k \right) + \lambda^{h-t-1} G_t^{h-t}, \tag{7}$$

where $G_t^k = \left( \sum_{l=0}^{k-1} \gamma^l r_{t+l} \right) + \gamma^k V_\phi(\mathbf{s}_{t+k})$ is the $k$-step return from time $t$. The estimated value $\tilde{V}(\mathbf{s})$ is treated as constant during critic training, as in regular actor-critic RL methods. In other words, the gradient of Eq. 6 does not flow through the states and actions in Eq. 7.

We further utilize the target value function technique (Mnih et al., 2015) to stabilize the training by smoothly transitioning from the previous value function to the newly fitted one, and use the target value function $V_{\phi'}$ to compute the policy loss (Eq. 5) and to estimate state values (Eq. 7). In addition, we apply observation normalization as is common in RL algorithms, which normalizes the state observation by a running mean and standard deviation calculated from the state observations in previous learning episodes. The pseudo code of our method is provided in Algorithm 1.

Our actor-critic formulation has several advantages that enable it to leverage simulation gradients effectively and efficiently. First, the terminal value function absorbs noisy landscape over long dynamics horizons and discontinuity introduced by early termination into a smooth function, as shown in Figure 2 (Right). This smooth surrogate formulation helps reduce the number of local spikes and alleviates the problem of easily getting stuck in local optima. Second, the short-horizon episodes avoid numerical problems when backpropagating the gradient through deeply nested update chains. Finally, the use of short-horizon episodes allows us to update the actor more frequently, which, when combined with parallel differentiable simulation, results in a significant speed up of training time.

## 4 EXPERIMENTS

We design experiments to investigate five questions: (1) How does our method compare to the state-of-the-art RL algorithms on classical RL control tasks, in terms of both sample efficiency and wall-clock time efficiency? (2) How does our method compare to the previous differentiable simulation-based policy learning methods? (3) Does our method scale to high-dimensional problems? (4) Is the terminal critic necessary? (5) How important is the choice of short horizon length $h$ for our method?

### 4.1 EXPERIMENT SETUP

To ensure a fair comparison for wall-clock time performance, we run all algorithms on the same GPU model (TITAN X) and CPU model (Intel Xeon(R) E5-2620). Furthermore, we conduct hyperparameter searches for all algorithms and report the performance of the best hyperparameters for each problem. In addition, we report the performance averaged from five individual runs for each algorithm on each problem. The details of the experimental setup are provided in Appendix A.2. We also experiment our method with a fixed hyperparameter setting and with a deterministic policy choice. Due to the space limit, the details of those experiments are provided in Appendix A.4.

### 4.2 BENCHMARK CONTROL PROBLEMS

For comprehensive evaluations, we select six broad control tasks, including five classical RL tasks across different complexity levels, as well as one high-dimensional control task with a large action

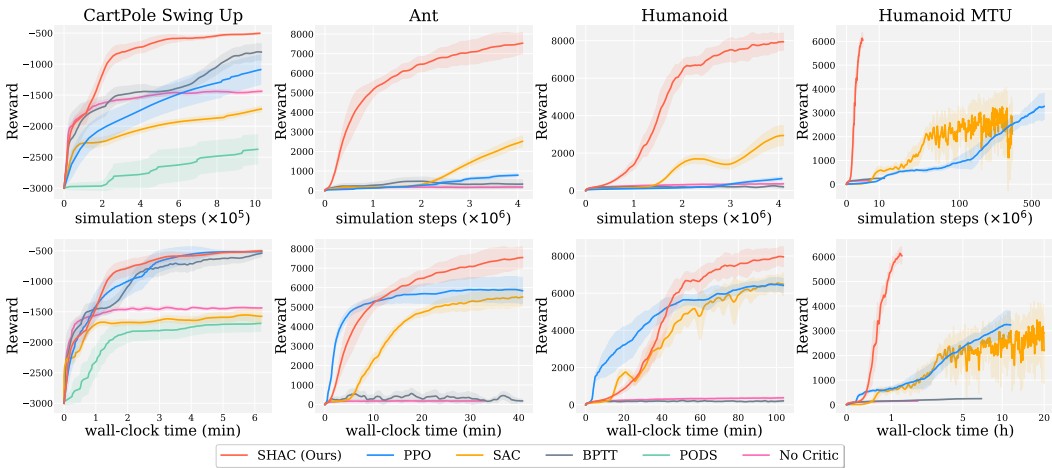

Figure 4: **Learning curves comparison on four benchmark problems.** Each column corresponds to a particular problem, with the top plot evaluating sample efficiency and the bottom plot evaluating wall-clock time efficiency. For better visualization, we truncate all the curves up to the maximal simulation steps/wall-clock time of our method (except for Humanoid MTU), and we provide the full plots in Appendix A.4. Each curve is averaged from five random seeds, and the shaded area shows the standard deviation. SHAC is more sample efficient than all baselines. Model-free baselines are competitive on wall-clock time on pedagogical environments such as the cartpole, but are much less effective as the problem complexity scales.

space. All tasks have stochastic initial states to further improve the robustness of the learned policy. We introduce four representative tasks in the main paper, and leave the others in the Appendix.

**Classical Tasks**: We select *CartPole Swing Up*, *Ant* and *Humanoid* as three representative RL tasks, as shown in Figure 1. Their difficulty spans from the simplest contact-free dynamics (*CartPole Swing Up*), to complex contact-rich dynamics (*Humanoid*). For *CartPole Swing Up*, we use $H = 240$ as the task horizon, whereas the other tasks use horizons of $H = 1000$.

**Humanoid MTU**: To assess how our method scales to high-dimensional tasks, we examine the challenging problem of muscle-actuated humanoid control (Figure 1, Right). We use the lower body of the humanoid model from Lee et al. (2019), which contains 152 muscle-tendon units (MTUs). Each MTU contributes one actuated degree of freedom that controls the contractile force applied to the attachment sites on the connected bodies. The task horizon for this problem is $H = 1000$.

To be compatible with differentiable simulation, the reward formulations of each problem are defined as differentiable functions. The details of each task are provided in Appendix A.3.

## 4.3 RESULTS

**Comparison to model-free RL.** We compare SHAC with Proximal Policy Optimization (PPO) (Schulman et al., 2017) (on-policy) & Soft Actor-Critic (SAC) (Haarnoja et al., 2018) (off-policy). We use high-performance implementations from *RL games* (Makoviichuk & Makoviychuk, 2021). To achieve state-of-the-art performance, we follow Makoviychuk et al. (2021): all simulation, reward and observation data remain on the GPU and are shared as PyTorch tensors between the RL algorithm and simulator. The PPO and SAC implementations are parallelized and operate on vectorized states and actions. With PPO we used short episode lengths, an adaptive learning rate, and large mini-batches during training to achieve the best possible performance.

As shown in the first row of Figure 4, our method shows significant improvements in sample efficiency over PPO and SAC in three classical RL problems, especially when the dimension of the problem increases (*e.g.*, *Humanoid*). The analytical gradients provided by the differentiable simulation allow us to efficiently acquire the expected policy gradient through a small number of samples. In contrast, PPO and SAC have to collect many Monte-Carlo samples to estimate the policy gradient.

Model-free algorithms typically have a lower per-iteration cost than methods based on differentiable simulation; thus, it makes sense to also evaluate wall-clock time efficiency instead of sample-efficiency alone. As shown in the second row of Figure 4, the wall-clock time performance of PPO, SAC, and our method are much closer than the sample efficiency plot. Interestingly, the training speed of our method is slower than PPO at the start of training. We hypothesize that the target value

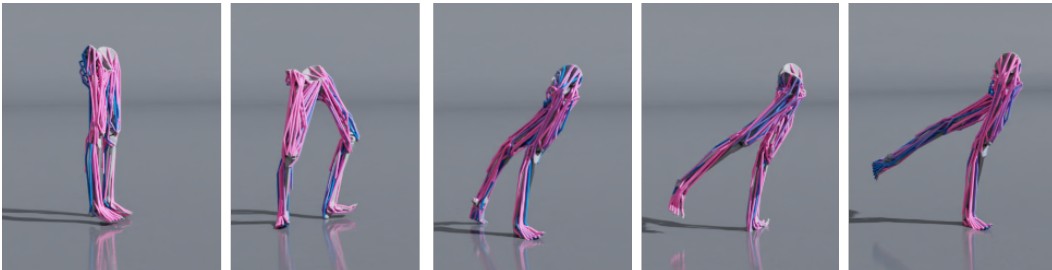

Figure 5: **Humanoid MTU**: A sequence of frames from a learned running gait. The muscle unit color indicates the activation level at the current state.

network in our method is initially requiring sufficient episodes to warm up. We also note that the backward time for our simulation is consistently around $2\times$ that of the forward pass. This indicates that our method still has room to improve its overall wall-clock time efficiency through the development of more optimized differentiable simulators with fast backward gradient computation. We provide a detailed timing breakdown of our method in Appendix A.4.

We observe that our method achieves better policies than RL methods in all problems. We hypothesize that, while RL methods are effective at exploration far from the solution, they struggle to accurately estimate the policy gradient near the optimum, especially in complex problems. We also compare our method to another model-free method REDQ (Chen et al., 2021b) in Appendix A.4.

**Comparison with previous gradient-based methods.** We compare our approach to three gradient-based learning methods: (1) Backpropagation Through Time (BPTT), which has been widely used in the differentiable simulation literature (Hu et al., 2019; Du et al., 2021), (2) PODS (Mora et al., 2021), and (3) Sample Enhanced Model-Based Policy Optimization (SE-MBPO) (Qiao et al., 2021).

BPTT: The original BPTT method backpropagates gradients over the entire trajectory, which results in exploding gradients as shown in Section 3.2. We modify BPTT to work on a shorter window of the tasks (64 steps for *CartPole* and 128 steps for other tasks), and also leverage parallel differentiable simulation to sample multiple trajectories concurrently to improve its time efficiency. As shown in Figure 4, BPTT successfully optimizes the policy for the contact-free *CartPole Swing Up* task, whereas it falls into local minima quickly in all other tasks involving contact. For example, the policy that BPTT learns for Ant is a stationary position leaning forward, which is a local minimum.

PODS: We compare to the first-order version of PODS, as the second-order version requires the full Jacobian of the state with respect to the whole action trajectory, which is not efficiently available in a reverse-mode differentiable simulator (including ours). Since PODS relies on a trajectory optimization step to optimize an open-loop action sequence, it is not clear how to accommodate early termination where the trajectory length can vary during optimization. Therefore, we test PODS performance only on the *CartPole Swing Up* problem. As shown in Figure 4, PODS quickly converges to a local optimum and is unable to improve further. This is because PODS is designed to be a method with high gradient exploitation but little exploration. Specifically, the line search applied in the trajectory optimization stage helps it converge quickly, but also prevents it from exploring more surrounding space. Furthermore, the extra simulation calls introduced by the line search and the slow imitation learning stage make it less competitive in either sample or wall-clock time efficiency.

SE-MBPO: Qiao et al. (2021) propose to improve a model-based RL method MBPO (Janner et al., 2019) by augmenting the rollout samples using data augmentation that relies on the Jacobian from the differentiable simulator. Although SE-MBPO shows high sample efficiency, the underlying model-based RL algorithm and off-policy training lead to a higher wall-clock time. As a comparison, the officially released code for SE-MBPO takes 8 hours to achieve a reasonable policy in the *Ant* problem used by Qiao et al. (2021), whereas our algorithm takes less than 15 minutes to acquire a policy with the same gait level in our *Ant* problem. Aiming for a more fair comparison, we adapt their implementation to work on our *Ant* problem in our simulator. However, we found that it could not successfully optimize the policy even after considerable hyperparameter tuning. Regardless, the difference in wall-clock time between two algorithms is obvious, and the training time of SE-MBPO is unlikely to be improved significantly by integrating it into our simulation environment. Furthermore, as suggested by Qiao et al. (2021), SE-MBPO does not generalize well to other tasks, whereas our method can be successfully applied to various complexity levels of tasks.

**Scalability to high-dimensional problems.** We test our algorithm and RL baselines on the *Humanoid MTU* example to compare their scalability to high-dimensional problems. With the large 152-dimensional action space, both PPO and SAC struggle to learn the policy as shown in Figure 4 (Right). Specifically, PPO and SAC learn significantly worse policies after more than 10 hours of training and with hundreds of millions of samples. This is because the amount of data required to accurately estimate the policy gradient significantly increases as the state and action spaces become large. In contrast, our method scales well due to direct access to the more accurate gradients provided by the differentiable simulation with the reparameterization techniques. To achieve the same reward level as PPO, our approach only takes around 35 minutes of training and 1.7M simulation steps. This results in over $17\times$ and $30\times$ wall-clock time improvement over PPO and SAC, respectively, and $382\times$ and $170\times$ more sample efficiency. Furthermore, after training for only 1.5 hours, our method is able to find a policy that has twice the reward of the best-performing policy from the RL methods. A learned running gait is visualized in Figure 5. Such scalability to high-dimensional control problems opens up new possibilities for applying differentiable simulation in computer animation, where complex character models are widely used to provide more natural motion.

**Ablation study on the terminal critic.** We introduce a terminal critic value in Eq. 5 to account for the long-term performance of the policy after the short episode horizon. In this experiment, we evaluate the importance of this term. By removing the terminal critic from Eq. 5, we get an algorithmic equivalent to BPTT with a short-horizon window and discounted reward calculation. We apply this no-critic variation on all four problems and plot the training curve in Figure 4, denoted by "No Critic". Without a terminal critic function, the algorithm is not able to learn a reasonable policy, as it only optimizes a short-horizon reward of the policy regardless of its long-term behavior.

**Study of short horizon length** $h$**.** The choice of horizon length $h$ is important for the performance of our method. $h$ cannot be too small, as it will result in worse value estimation by td-$\lambda$ (Eq. 7) and underutilize the power of the differentiable simulator to predict the sensitivity of future performance to the policy weights. On the other hand, a horizon length that is too long will lead to a noisy optimization landscape and less-frequent policy updates. Empirically, we find that a short horizon length $h = 32$ with $N = 64$ parallel trajectories works well for all tasks in our experiments. We conduct a study of short horizon length on the *Ant* task to show the influence of this hyperparameter. We run our algorithm with six short horizon lengths $h = 4, 8, 16, 32, 64, 128$. We set the corresponding number of parallel trajectories $N = 512, 256, 128, 64, 32, 16$ for the variant, such that each one generates the same amount of samples in single learning episode. We run each variant for the same number of episodes $M = 2000$ with 5 individual random seeds. In Figure 6, we report the average reward of the

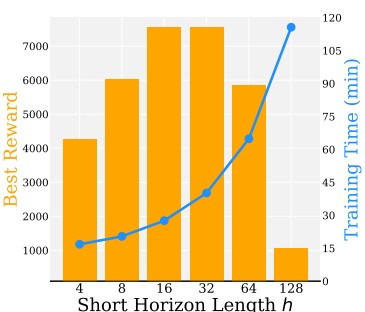

Figure 6: **Study of short horizon length** $h$ **on Ant problem**. A small $h$ results in worse value estimation. A too large $h$ leads to an ill-posed optimization landscape and longer training time.

best policies from 5 runs for each variant, as well as the total training time. As expected, the best reward is achieved when $h = 16$ or 32, and the training time scales linearly as $h$ increases.

## 5 CONCLUSION AND FUTURE WORK

In this work, we propose an approach to effectively leverage differentiable simulation for policy learning. At the core is the use of a critic network that acts as a smooth surrogate to approximate the underlying noisy optimization landscape. In addition, a truncated learning window is adopted to alleviate the problem of exploding/vanishing gradients during deep backward paths. Equipped with the developed parallel differentiable simulation, our method shows significantly higher sample efficiency and wall-clock time efficiency over state-of-the-art RL and gradient-based methods, especially when the problem complexity increases. As shown in our experiments, model-free methods demonstrate efficient learning at the start of training, but SHAC is able to achieve superior performance after a sufficient number of episodes. A compelling future direction for research is how to combine model-free methods with our gradient-based method in order to leverage the strengths of both. Furthermore, in our method, we use a fixed and predetermined short horizon length $h$ throughout the learning process; however, future work may focus on implementing an adaptive short horizon schedule that varies with the status of the optimization landscape.

## ACKNOWLEDGEMENTS

We thank the anonymous reviewers for their helpful comments in revising the paper. The majority of this work was done during the internship of Jie Xu at NVIDIA, and we thank Tae-Yong Kim for his mentorship. This work is also partially supported by Defense Advanced Research Projects Agency (FA8750-20-C-0075).

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

# A  APPENDIX

## A.1  POLICY LOSS GRADIENT COMPUTATION

We minimize the following policy loss to improve the policy network $\pi_\theta$:

$$\mathcal{L}_\theta = -\frac{1}{Nh} \sum_{i=1}^{N} \left[ \left( \sum_{t=t_0}^{t_0+h-1} \gamma^{t-t_0} \mathcal{R}(\mathbf{s}_t^i, \mathbf{a}_t^i) \right) + \gamma^h V_\phi(\mathbf{s}_{t_0+h}^i) \right], \tag{8}$$

where $N$ is the number of trajectories, $h$ is the short horizon length, $t_0$ is the starting time of each trajectory, $\mathbf{s}_t^i$ and $\mathbf{a}_t^i$ are the state and actions at time step $t$ of trajectory $\tau_i$.

To compute its gradient, we treat the differentiable simulator as a differentiable layer (with backward pass shown in Eq. 1) in the computation graph for policy loss $\mathcal{L}_\theta$, and acquiring the gradients $\frac{\partial \mathcal{L}_\theta}{\partial \theta}$ by PyTorch with its reverse-mode computation. Mathematically, we have the following terminal adjoints for each trajectory $\tau_i$

$$\frac{\partial \mathcal{L}_\theta}{\partial \mathbf{s}_{t_0+h}^i} = -\gamma^h \frac{1}{Nh} \frac{\partial V_\phi(\mathbf{s}_{t_0+h}^i)}{\partial \mathbf{s}_{t_0+h}^i} \tag{9}$$

From the last time step $t_0 + h$, we can compute the adjoints in the previous steps $t_0 \leq t < t_0 + h$ in reverse order:

$$\frac{\partial \mathcal{L}_\theta}{\partial \mathbf{s}_t^i} = -\gamma^{t-t_0} \frac{1}{Nh} \frac{\partial \mathcal{R}(\mathbf{s}_t^i, \mathbf{a}_t^i)}{\partial \mathbf{s}_t^i} + \left( \frac{\partial \mathcal{L}_\theta}{\partial \mathbf{s}_{t+1}^i} \right) \left( \left( \frac{\partial \mathcal{F}}{\partial \mathbf{s}_t^i} \right) + \left( \frac{\partial \mathcal{F}}{\partial \mathbf{a}_t^i} \right) \left( \frac{\partial \pi_\theta(\mathbf{s}_t^i)}{\partial \mathbf{s}_t^i} \right) \right) \tag{10}$$

$$\frac{\partial \mathcal{L}_\theta}{\partial \mathbf{a}_t^i} = -\gamma^{t-t_0} \frac{1}{Nh} \frac{\partial \mathcal{R}(\mathbf{s}_t^i, \mathbf{a}_t^i)}{\partial \mathbf{a}_t^i} + \left( \frac{\partial \mathcal{L}_\theta}{\partial \mathbf{s}_{t+1}^i} \right) \left( \frac{\partial \mathcal{F}}{\partial \mathbf{a}_t^i} \right) \tag{11}$$

From all the computed adjoints, we can compute the policy loss gradient by

$$\frac{\partial \mathcal{L}_\theta}{\partial \theta} = \sum_{i=1}^{N} \sum_{t=t_0}^{t_0+h-1} \left( \frac{\partial \mathcal{L}_\theta}{\partial \mathbf{a}_t^i} \right) \left( \frac{\partial \pi_\theta(\mathbf{s}_t^i)}{\partial \theta} \right) \tag{12}$$

To be noted, for our stochastic policy $\mathbf{a}_t \sim \mathcal{N}(\mu_\theta(\mathbf{s}_t), \sigma_\theta(\mathbf{s}_t))$, we use the reparamterization sampling method, which allows us to compute the gradients $\frac{\partial \pi_\theta(\mathbf{s})}{\partial \theta}$ and $\frac{\partial \pi_\theta(\mathbf{s})}{\partial \mathbf{s}_t}$.

## A.2  EXPERIMENT SETUP

To ensure a fair comparison for wall-clock time performance, we run all algorithms on the same GPU model (TITAN X) and CPU model (Intel Xeon(R) E5-2620). Furthermore, we conduct extensive hyperparameter searches for all algorithms and report the performances of the best hyperparameter settings on each problem. For our method, we run for 500 learning episodes for *CartPole Swing Up* problem and for 2000 learning episodes for other five problems. For baseline algorithms, we run each of them for sufficiently long time in order to acquire the policies with the highest rewards. We run each algorithm with five individual random seeds, and report the average performance.

### A.2.1  BASELINE ALGORITHM IMPLEMENTATIONS

**Policy and Value Network Structures** We use multilayer perceptron (MLP) as the policy and value network structures for all algorithms. We use ELU as our activation function for the hidden layers and apply layer normalization to each hidden layer. Since different problems have different state and action dimensionalities, we use different network sizes for different tasks.

**Codebase and Implementations** For PPO and SAC, we use high-performance implementations of both methods available in RL games (Makoviichuk & Makoviychuk, 2021). To achieve state-of-the-art performance we follow the approach proposed by Makoviychuk et al. (2021) where all the simulation, reward and observation data stays on GPU and is shared as PyTorch tensors between the RL algorithm and the parallel simulator. Both PPO and SAC implementations are parallelized and operate on vectorized states and actions. With PPO we used short episode lengths, an adaptive learning rate, and large mini-batch sizes during training to achieve the maximum possible performance.

For BPTT, the original BPTT requires to backprapagate gradients over the entire trajectory and results in exploding gradients problem. In order to make it work on our long-horizon tasks with

complex dynamics, we modify BPTT to work on a shorter window of the tasks (64 steps for *CartPole* and 128 steps for other tasks). We further improve its wall-clock time efficiency by leveraging parallel differentiable simulation to sample multiple trajectories concurrently.

For PODS, due to the lack of released code, we implement our own version of it. Specifically, we implement the first-order version of PODS proposed by Mora et al. (2021) since the second-order version requires the full Jacobian of the state with respect to the whole action trajectory, which is typically not efficiently available in a reverse-mode differentiable simulator (including ours).

For SE-MBPO, we adapt the implementation released by Qiao et al. (2021) into our benchmark problem.

### A.2.2    IMPLEMENTATION DETAILS OF OUR METHOD

**Policy and Value Network Structures** We use multilayer perceptron (MLP) as the policy for our method. We use ELU as our activation function for the hidden layers and apply layer normalization to each hidden layer. Since different problems have different state and action dimensionalities, we use different network sizes for different tasks. Basically, we use larger policy and value network structures for problems with higher degrees of freedoms. Empirically we find the performance of our method is insensitive to the size of the network structures. We report the network sizes that are used in our experiments for Figure 4 in Table 1 for better reproducibility. For other problems (*i.e.*, HalfCheetah and Hopper), refer to our released code for the adopted hyperparameters.

Table 1: The network setting of our method on each problem.

| Problems | Policy Network Structure | Value Network Structure |
|---|---|---|
| **CartPole Swing Up** | [64, 64] | [64, 64] |
| **Ant** | [128, 64, 32] | [64, 64] |
| **Humanoid** | [256, 128] | [128, 128] |
| **Humanoid MTU** | [512, 256] | [256, 256] |

### A.2.3    HYPERPARAMETERS SETTINGS

Since our benchmark problems span across a wide range of complexity and problem dimentionality, the optimal hyperparameters in expected will be different across problems. We conduct an extensive hyperparameter search for all algorithms, especially for PPO and SAC, and report their best performing hyperparameter settings for fair comparison. For PPO, we adopt the adaptive learning rate strategy implemented in RL games (Makoviichuk & Makoviychuk, 2021). The hyperparameters of PPO and SAC we used in the experiments are reported in Table 2 and 3.

Table 2: The hyperparameter setting of PPO on each problem.

| Parameter names | CartPole | Ant | Humanoid | Humanoid MTU |
|---|---|---|---|---|
| short horizon length $h$ | 240 | 32 | 32 | 32 |
| number of parallel environments $N$ | 32 | 2048 | 1024 | 1024 |
| discount factor $\gamma$ | 0.99 | 0.99 | 0.99 | 0.99 |
| GAE $\lambda$ | 0.95 | 0.95 | 0.95 | 0.95 |
| minibatch size | 1920 | 16384 | 8192 | 8192 |
| mini epochs | 5 | 5 | 5 | 6 |

For our method, we also conduct hyperparameter searches for each problem, including the short horizon length $h$, the number of parallel environments $N$, and the policy and critic learning rates. The learning rates follow a *linear decay* schedule over episodes. We report performance from the

Table 3: The hyperparameter setting of SAC on each problem.

| Parameter names | CartPole | Ant | Humanoid | Humanoid MTU |
|---|---|---|---|---|
| num steps per episode $h$ | 128 | 128 | 128 | 128 |
| number of parallel environments $N$ | 32 | 128 | 64 | 256 |
| discount factor $\gamma$ | 0.99 | 0.99 | 0.99 | 0.99 |
| $\alpha$ learning rate | 0.005 | 0.005 | 0.0002 | 0.0002 |
| actor learning rate | 0.0005 | 0.0005 | 0.0003 | 0.0003 |
| critic learning rate | 0.0005 | 0.0005 | 0.0003 | 0.0003 |
| critic $\tau$ | 0.005 | 0.005 | 0.005 | 0.005 |
| minibatch size | 1024 | 4096 | 2048 | 4096 |
| replay buffer size | 1e6 | | | |
| learnable temperature | True | | | |
| number of seed steps | 2 | | | |

best hyperparameter settings in Figure 4, and report the hyperparameters for each problem in Table 4.

Table 4: The optimal hyperparameter setting of our method on each problem.

| Parameter names | CartPole | Ant | Humanoid | Humanoid MTU |
|---|---|---|---|---|
| short horizon length $h$ | 32 | | | |
| number of parallel environments $N$ | 64 | | | |
| policy learning rate | 0.01 | 0.002 | | |
| critic learning rate | 0.001 | 0.002 | 0.0005 | |
| target value network $\alpha$ | 0.2 | | 0.995 | |
| discount factor $\gamma$ | 0.99 | | | |
| value estimation $\lambda$ | 0.95 | | | |
| Adam $(\beta_1, \beta_2)$ | $(0.7, 0.95)$ | | | |
| number of critic training iterations | 16 | | | |
| number of critic training minibatches | 4 | | | |
| number of episodes $M$ | 500 | 2000 | | |

Although the optimal hyperparameters are different for different problems due to the different complexity of the problem and we report the best setting for fair comparison against baseline algorithms, in general, we found the set of hyperparameters in Table 5 works reasonably well on all problems (leading to slightly slower convergence speed in CartPole Swing Up and Ant tasks due to large $\alpha$ for target value network). More analysis is provided in Appendix A.4.3.

## A.3 BENCHMARK CONTROL PROBLEMS

We describe more details of our benchmark control problems in this section. We evaluate our algorithm and baseline algorithms on six benchmark tasks, including five classical RL benchmark problems and one high-dimensional problem. We define the reward functions to be differentiable functions in order to incorporate them into our differentiable simulation framework.

### A.3.1 CARTPOLE SWING UP

CartPole Swing Up is one of the simplest classical RL control tasks. In this problem, the control policy has to swing up the pole to the upward direction and keeps it in that direction as long as possible. We have 5-dimensional observation space as shown in Table 6 and 1-dimensional action to control the torque applied to the prismatic joint of the cart base.

The single-step reward is defined as:
$$\mathcal{R} = -\theta^2 - 0.1\dot{\theta}^2 - 0.05x^2 - 0.1\dot{x}^2 \tag{13}$$

Table 5: A general setting of hyperparameters of our method.

| Hyperparameter names | Values |
|---|---|
| short horizon length $h$ | 32 |
| number of parallel environments $N$ | 64 |
| policy learning rate | 0.002 |
| critic learning rate | 0.0005 |
| discount factor $\gamma$ | 0.99 |
| value estimation $\lambda$ | 0.95 |
| target value network $\alpha$ | 0.995 |
| number of critic training iterations | 16 |
| number of critic training minibatches | 4 |

Table 6: Observation vector of CartPole Swing Up problem

| Observation | Degrees of Freedom |
|---|---|
| position of cart base: $x$ | 1 |
| velocity of the cart base: $\dot{x}$ | 1 |
| sine and cosine of the pole angle: $\sin(\theta), \cos(\theta)$ | 2 |
| angular velocity of pole: $\dot{\theta}$ | 1 |

The initial state is randomly sampled. The task horizon is 240 steps and there is no early termination in this environment so that it can be used to test the algorithms which are not compatible with early termination strategy.

### A.3.2 HALFCHEETAH

In this problem, a two-legged cheetah robot is controlled to run forward as fast as possible. We have 17-dimensional observation space as shown in Table 7 and 6-dimensional action to control the torque applied to each joint.

Table 7: Observation vector of HalfCheetah problem

| Observation | Degrees of Freedom |
|---|---|
| height of the base: $h$ | 1 |
| rotation angle of the base | 1 |
| linear velocity of the base: $v$ | 2 |
| angular velocity of the base | 1 |
| joint angles | 6 |
| joint angle velocities | 6 |

The single-step reward is defined as:

$$\mathcal{R} = v_x - 0.1\|a\|^2, \tag{14}$$

where $v_x$ is the forward velocity.

The initial state is randomly sampled. The task horizon is 1000 steps and there is no early termination in this environment.

### A.3.3 HOPPER

In this problem, a jumping hopper is controlled to run forward as fast as possible. We have 11-dimensional observation space as shown in Table 8 and 3-dimensional action to control the torque applied to each joint.

Table 8: Observation vector of Hopper problem

| Observation | Degrees of Freedom |
| --- | --- |
| height of the base: $h$ | 1 |
| rotation angle of the base $\theta$ | 1 |
| linear velocity of the base: $v$ | 2 |
| angular velocity of the base | 1 |
| joint angles | 3 |
| joint angle velocities | 3 |

The single-step reward is defined as:
$$\mathcal{R} = \mathcal{R}_v + \mathcal{R}_{height} + \mathcal{R}_{angle} - 0.1\|a\|^2, \tag{15}$$
where $\mathcal{R}_v = v_x$ is the forward velocity, $\mathcal{R}_{height}$ is designed to penalize the low height state, defined by:
$$\mathcal{R}_{height} = \begin{cases} -200\Delta_h^2, & \Delta_h < 0 \\ \Delta_h, & \Delta_h \geq 0 \end{cases} \tag{16}$$
$$\Delta_h = \text{clip}(h + 0.3, -1, 0.3) \tag{17}$$
and the $\mathcal{R}_{angle}$ is designed to encourage the upper body of the hopper to be as upward as possible, defined by
$$\mathcal{R}_{angle} = 1 - \left(\frac{\theta}{30°}\right)^2 \tag{18}$$

The initial state is randomly sampled. The task horizon is 1000 steps and early termination is triggered when the height of the hopper is lower than $-0.45$m.

### A.3.4 ANT

In this problem, a four-legged ant is controlled to run forward as fast as possible. We have 37-dimensional observation space as shown in Table 9 and 8-dimensional action to control the torque applied to each joint.

The single-step reward is defined as:
$$\mathcal{R} = \mathcal{R}_v + 0.1\mathcal{R}_{up} + \mathcal{R}_{heading} + \mathcal{R}_{height}, \tag{19}$$
where $\mathcal{R}_v = v_x$ is the forward velocity, $\mathcal{R}_{up} = \text{projection(upward direction)}$ encourages the agent to be vertically stable, $\mathcal{R}_{heading} = \text{projection(forward direction)}$ encourages the agent to run straight forward, and $\mathcal{R}_{height} = h - 0.27$ is the height reward.

The initial state is randomly sampled. The task horizon is 1000 steps and early termination is triggered when the height of the ant is lower than $0.27$m.

### A.3.5 HUMANOID

In this problem, a humanoid robot is controlled to run forward as fast as possible. We have a 76-dimensional observation space as shown in Table 10 and a 21-dimensional action vector to control the torque applied to each joint.

Table 9: Observation vector of Ant problem

| Observation | Degrees of Freedom |
| --- | --- |
| height of the base: $h$ | 1 |
| rotation quaternion of the base | 4 |
| linear velocity of the base: $v$ | 3 |
| angular velocity of the base | 3 |
| joint angles | 8 |
| joint angle velocities | 8 |
| up and heading vectors projections | 2 |
| actions in last time step | 8 |

Table 10: Observation vector of Humanoid problem

| Observation | Degrees of Freedom |
| --- | --- |
| height of the torso: $h$ | 1 |
| rotation quaternion of the torso | 4 |
| linear velocity of the torso: $v$ | 3 |
| angular velocity of the torso | 3 |
| joint angles | 21 |
| joint angle velocities | 21 |
| up and heading vectors projections | 2 |
| actions in last time step | 21 |

The single-step reward is defined as:

$$\mathcal{R} = \mathcal{R}_v + 0.1\mathcal{R}_{up} + \mathcal{R}_{heading} + \mathcal{R}_{height} - 0.002\|\mathbf{a}\|^2, \tag{20}$$

where $\mathcal{R}_v = v_x$ is the forward velocity, $\mathcal{R}_{up}$ is the projection of torso on the upward direction encouraging the agent to be vertically stable, $\mathcal{R}_{heading}$ is the projection of torso on the forward direction encouraging the agent to run straight forward, and $\mathcal{R}_{height}$ is defined by:

$$\mathcal{R}_{height} = \begin{cases} -200\Delta_h^2, & \Delta_h < 0 \\ 10\Delta_h, & \Delta_h \geq 0 \end{cases} \tag{21}$$

$$\Delta_h = \text{clip}(h - 0.84, -1, 0.1) \tag{22}$$

The initial state is randomly sampled. The task horizon is 1000 steps and early termination is triggered when the height of torso is lower than 0.74m.

### A.3.6 HUMANOID MTU

This is the most complex problem designed to assess the scalability of the algorithms. In this problem, a lower body of the humanoid model from Lee et al. (2019) is actuated by 152 muscle-tendon units (MTUs). Each MTU contributes one actation degree of freedom that controls the contractile force applied to the attachment sites on the connected bodies. We have a 53-dimensional observation space as shown in Table 11 and a 152-dimensional action vector.

The single-step reward is defined as:

$$\mathcal{R} = \mathcal{R}_v + 0.1\mathcal{R}_{up} + \mathcal{R}_{heading} + \mathcal{R}_{height} - 0.001\|\mathbf{a}\|^2, \tag{23}$$

where $\mathcal{R}_v = v_x$ is the forward velocity, $\mathcal{R}_{up} = \text{projection(upward direction)}$ encourages the agent to be vertically stable, $\mathcal{R}_{heading} = \text{projection(forward direction)}$ encourages the agent to run straight

Table 11: Observation vector of Humanoid MTU problem

| Observation | Degrees of Freedom |
|---|---|
| height of the pelvis: $h$ | 1 |
| rotation quaternion of the pelvis | 4 |
| linear velocity of the pelvis: $v$ | 3 |
| angular velocity of the pelvis | 3 |
| joint angles | 22 |
| joint angle velocities | 18 |
| up and heading vectors projections | 2 |

forward, and $\mathcal{R}_{height}$ is defined by:

$$\mathcal{R}_{height} = \begin{cases} -200\Delta_h^2, & \Delta_h < 0 \\ 4\Delta_h, & \Delta_h \geq 0 \end{cases} \tag{24}$$

$$\Delta_h = \text{clip}(h - 0.51, -1, 0.05) \tag{25}$$

The initial state is randomly sampled. The task horizon is 1000 steps and early termination is triggered when the height of pelvis is lower than $0.46$m.

## A.4 MORE RESULTS

### A.4.1 FULL LEARNING CURVE COMPARISON

In Section 4.3, we provide the truncated version of the learning curves in Figure 4 for better visualization. We provide the full version of the learning curves in Figure 7, where the curve for each algorithm continues to the end of the training of that algorithm. In addition, we include the experiment results for the other two classic RL environments, *HalfCheetah* and *Hopper*. From the figure, we can see that our method shows extreme improvements in sample efficiency over other methods and constantly achieves better policies given the same amount of training time.

### A.4.2 WALL-CLOCK TIME BREAKDOWN OF TRAINING

We provide the detailed wall-clock time performance breakdown of a single learning episode of our method in Table 12. Since we care about the wall-clock time performance of the algorithm, we use this table for a better analysis of which is the time bottleneck in our algorithm. The performance is measured on a desktop with GPU model TITAN X and CPU model Intel Xeon(R) E5-2620 @ 2.10GHz. As shown in the table, the forward simulation and backward simulation time scales up when the dimensionality of the problem increases, while the time spent in critic value function training is almost constant across problems. As expected, the differentiable simulation brings extra overhead for its backward gradient computation. Specifically in our differentiable simulation, the backward computation time is roughly $2\times$ of the forward time. This indicates that our method still has room to improve its overall wall-clock time efficiency through the development of more optimized differentiable simulators with fast backward computation.

### A.4.3 FIXED NETWORK ARCHITECTURE AND HYPERPARAMETERS

In the previous results, we fine tune the network architectures of policy and value function for each algorithm on each problem to get their maximal performance for a fair comparison. In this section, we test the robustness of our method by using a fixed network architecture and fixed set of hyperparameters (*e.g.* learning rates) to train on all problems. Specifically, we use the same architecture and the hyperparameters used in the *Humanoid* problem, and plot the training curves in Figure 8.

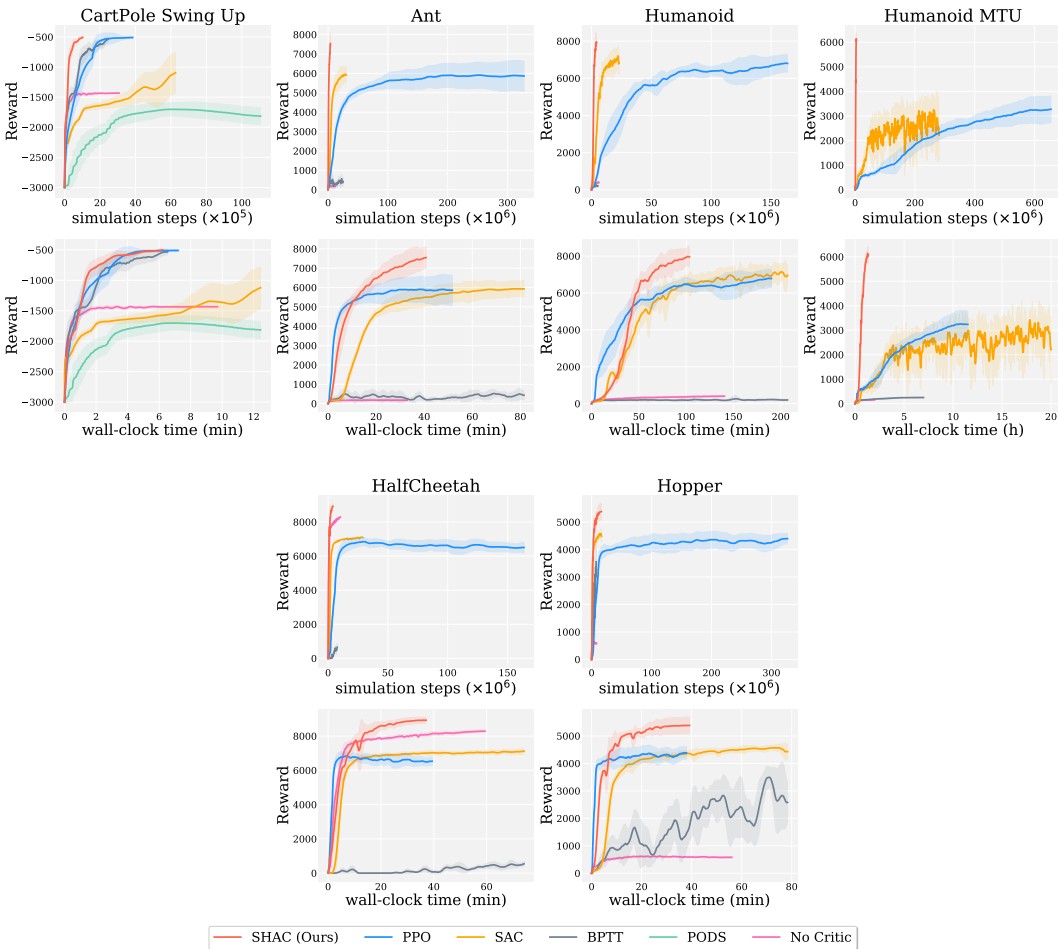

Figure 7: **Full learning curves of algorithms on six benchmark problems.** Each curve is averaged from five random seeds and the shaded area shows the standard deviation. Each problem is shown in a column, with the plot on the top showing the sample efficiency and the curves on the bottom for wall-clock time efficiency. We run our method for $500$ learning episodes on CartPole Swing Up, and for $2000$ learning episodes for other three tasks. We run other algorithms for sufficiently long time for a comprehensive comparison (either until convergence, or up to $2\times$ of the training time compared to our method on first three problems, and up to $20$ hours in Humanoid MTU).

|  | **Forward** (s) | **Backward** (s) | **Critic Training** (s) |
|---|---|---|---|
| **CartPole SwingUp** | 0.15 | 0.23 | 0.34 |
| **Ant** | 0.25 | 0.37 | 0.32 |
| **Humanoid** | 0.91 | 1.97 | 0.32 |
| **SNU Humanoid** | 0.82 | 1.66 | 0.33 |

Table 12: **Wall-clock performance breakdown of a single training episode.** The forward stage includes simulation, reward calculation, and observations. Backward includes the simulation gradient calculation and actor update. Critic training, which is specific to our method, is listed individually, and is generally a small proportion of the overall training time.

### A.4.4 DETERMINISTIC POLICY

Our method does not constrain the policy to be stochastic or deterministic. We choose to use the stochastic policy in the most experiments for the extra exploration that it provides. In this section, we experiment our method with the deterministic policy choice. Specifically, we change the policy of our method in each problem from stochastic policy to deterministic policy while keeping all

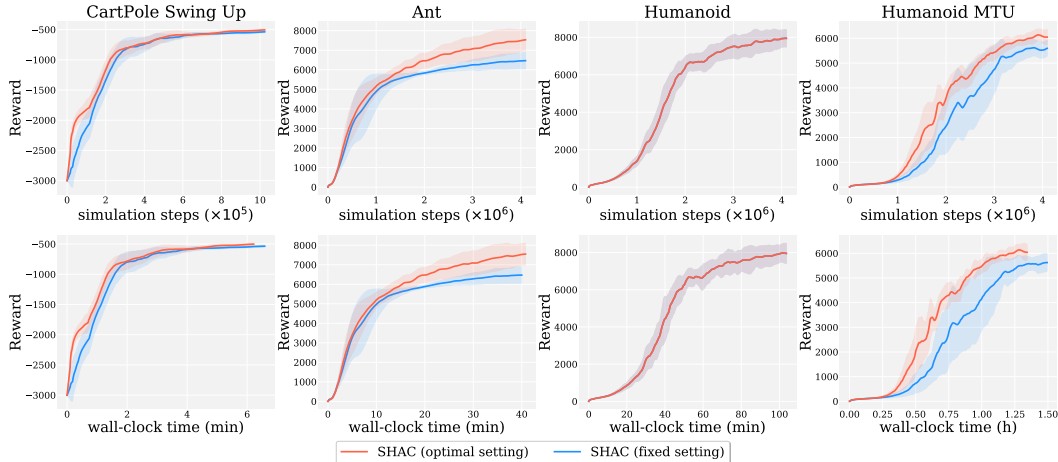

Figure 8: **Learning curves of our method with fixed network architectures and learning rates.** We use the same network architectures and learning rates used in *Humanoid* problem on all other problems, and plot the training curves comparison with the ones using optimal settings. The plot shows that our method still performs reasonably well with the fixed network and learning rates settings.

other hyperparameters such as network dimensions and learning rates the same as shown in Table 4. The training curves of the deterministic policy is plotted in Figure 9. The results show that our method works reasonably well with deterministic policy, and sometimes the deterministic policy even outperforms the stochastic policy (*e.g. Humanoid*). The small performance drop on the *Ant* problem comes from one single random seed (out of five) which results in a sub-optimal policy.

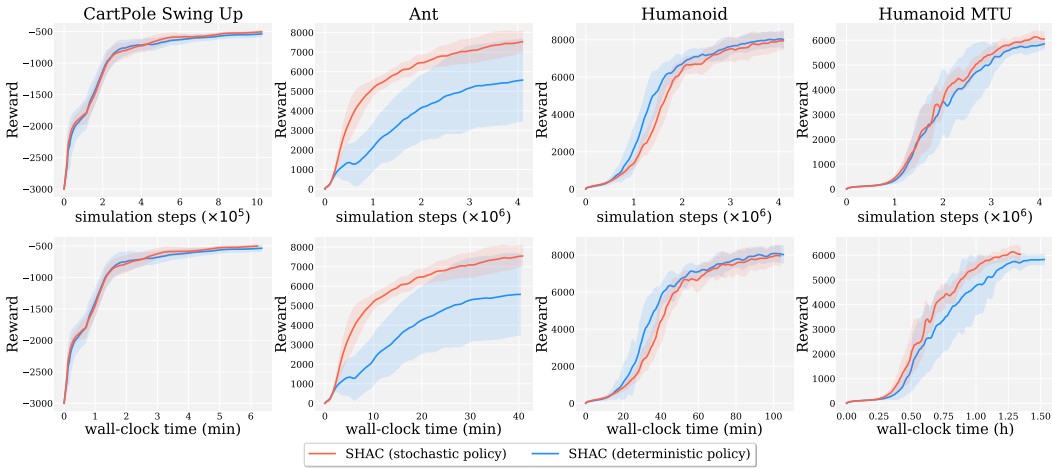

Figure 9: **Learning curves of our method with deterministic policy.** We test our method with deterministic policy choice. We use the same network sizes and the hyperparameters as used in the stochastic policy and remove the policy output stochasticity. We run our method on each problem with five individual random seeds. The results show that our method with deterministic policy works reasonably well on all problems, and sometimes the deterministic policy even outperforms the stochastic policy (*e.g., Humanoid*). The small performance drop on the *Ant* problem comes from one single seed (out of five) which results in a sub-optimal policy.

### A.4.5    COMPARISON TO REDQ

In this section, we compare our method to another advanced model-free reinforcement learning algorithm, namely REDQ (Chen et al., 2021b). REDQ is an off-policy method and is featured by its high sample efficiency. We experiment the officially released code of REDQ on four of our benchmark problems: *CartPole Swing Up, Ant, Humanoid,* and *Humanoid MTU*, and provide the comparison between our method and REDQ in Figure 10. From the plots, we see that REDQ

demonstrates its high sample efficiency in the *CartPole Swing Up* and *Ant* problems due to its off-policy updates and the randomized Q function ensembling strategy. However, REDQ is extremely slow in wall-clock training time. For example, in *Ant* problem, REDQ takes 27 hours to achieve 5000 rewards while SHAC only takes 13 minutes to achieve the same reward. Such huge difference in wall-clock time performance renders REDQ a less attractive method than SHAC in the settings where the policy is learned in a fast simulation. Furthermore, REDQ fails to train on our high-dimensional problems, *Humanoid* and *Humanoid MTU* problems.

### A.4.6 COMPARISON TO MODEL-BASED RL

In this section, we compare our method to a model-based reinforcement learning algorithm. In a typical model-based reinforcement learning algorithm, the learned model can be used in two ways: (1) data augmentation, (2) policy gradient estimation with backpropagation through a differentiable model.

In the data augmentation category, the learned model is used to provide rollouts around the observed environment transitions, as introduced in the DYNA algorithm (Sutton, 1990). The core idea being that the model can be used to provide additional training samples in the neighborhood of observed environment data. The core idea in DYNA was extended by STEVE (Buckman et al., 2018) and MBPO (Janner et al., 2019), which use interpolation between different horizon predictions compared to a single step roll-out in DYNA. However, the performance of such approaches rapidly degrades with increasing model error. Notably, as noted by Janner et al. (2019), empirically the one-step rollout is a very strong baseline to beat in part because error in model can undermine the advantage from model-based data-augmentation.

In contrast the model can be used for policy-value gradient estimation, which originates from Nguyen & Widrow (1990) and Jordan & Rumelhart (1992). This idea has been extended multiple times such as in PILCO (Deisenroth & Rasmussen, 2011), SVG (Heess et al., 2015), and MAAC (Clavera et al., 2020). Despite the core method being very similar to the original idea, the difference between recent methods such as SVG and MAAC emerges in the details of what data is used in estimation. SVG uses only real samples, while MAAC uses both real and model roll-outs. But most of these methods still rely on model error being low for policy and value computation.

While SHAC resembles these gradient-based methods, our model usage fundamentally differs from previous approaches. SHAC is the first method that combines actor-critic methods with analytical gradient computation from a differentiable simulator model. The analytical gradient solves the model learning issues which plague methods both in model-based data-augmentation and direct gradient estimators. At the same time, SHAC prevents exploding/vanishing gradients in BPTT through truncated roll-outs and a terminal critic function. SHAC offers improvement over prior model-based methods in both improved efficiency as well as scale-up to higher dimensional problems, beyond the ones reported in these methods.

To further illustrate the advantage of our method over model-based RL, we conduct the experiments to compare to Model-Based Policy Optimization (MBPO) (Janner et al., 2019). We run the officially released code of MBPO on four of our benchmark problems: *CartPole Swing Up, Ant, Humanoid,* and *Humanoid MTU*, and provide the comparison between our method and MBPO in Figure 10. As shown in the figure, the MBPO with the default parameters does not work well on our benchmark problems except *CartPole Swing Up*. We hypothesize that it is because of the hyperparameter sensitivity of model-based RL method. Although a fair comparision can be further conducted by tuning the hyperparameters of MBPO, an implicit comparison can be made through the comparison with REDQ. In REDQ paper (Chen et al., 2021b), it is reported that REDQ is up to 75% faster in wall-clock time than MBPO. This indirectly indicates that our method has significant advantages in the wall-clock time efficiency over MBPO.

### A.5 MORE OPTIMIZATION LANDSCAPE ANALYSIS

In Figure 2, we show the original landscape of the problem and the surrogate landscape defined by SHAC. However the smoothness of the surrogate landscape sometimes does not necessarily results in a smooth gradient when the evaluation trajectory is stochastic and the gradient is computed by averaging over policy stochasticity (Parmas et al., 2018). To further analyze the gradient of the landscape, we plot the comparison of the computed single-weight analytical gradient and the single-

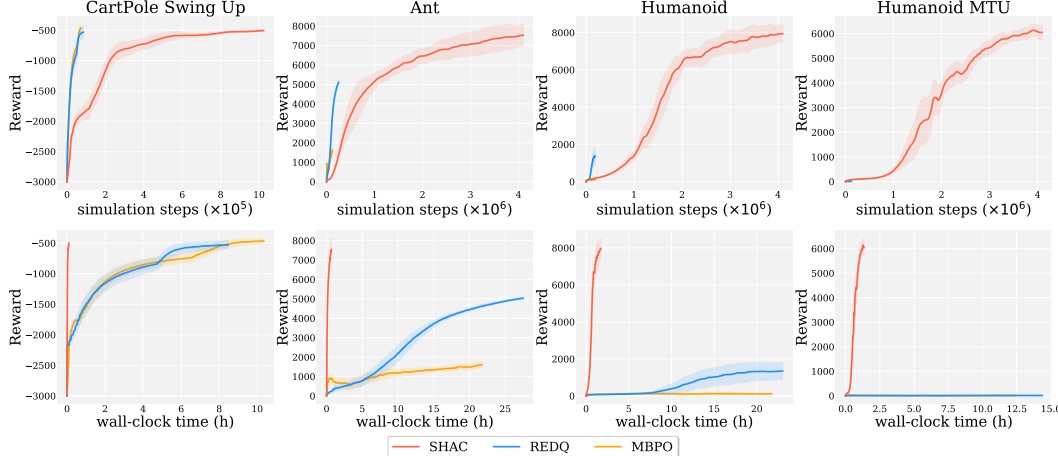

Figure 10: **Comparison with REDQ and MBPO.** We compare SHAC to REDQ and MBPO. While REDQ has very excellent sample efficiency, its wall-clock time efficiency is very poor ($102\times$ slower than SHAC in *CartPole Swing Up* and $162\times$ slower than SHAC in *Ant*), which renders it as a less attractive method than SHAC in the settings where the policy is learned in a fast simulation. The offical MBPO code works well on the *CartPole Swing Up* problem while works even worse than REDQ on other three tasks. Furthermore, both REDQ and MBPO fail to train on our high-dimensional problems, *Humanoid* and *Humanoid MTU*, with default hyperparameters.

weight gradient computed by finite difference of the landscape (shown in Figure 2 Right) in Figure 11. The plot demonstrates that our computed gradient is well-behaved and close enough to the finite difference gradient.

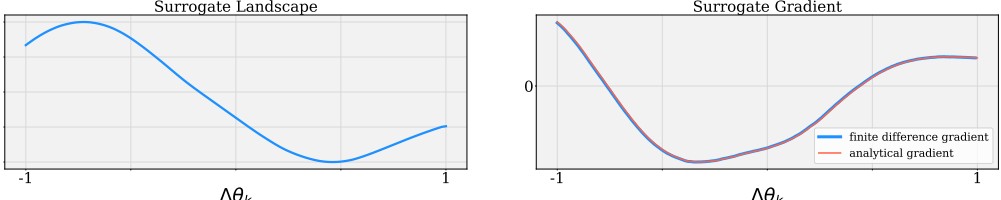

Figure 11: **Single-weight landscape and gradient of the SHAC loss.** In the left figure, we select one single weight from a policy and change its value by $\Delta\theta_k \in [-1, 1]$ to plot the task loss landscapes of SHAC w.r.t. one policy parameter (same as Figure 2 right). The short horizon length is $h = 32$. Each policy variant is evaluated by 128 stochastic trajectories and we set the same random seed before the evaluation of each policy variant. In the right figure, we plot the finite difference gradient and the analytical gradient along that single weight axis. It shows that the analytical gradient is well-behaved and close enough to the finite difference gradient.

Furthermore, we compare the gradient of the BPTT method and the gradient of SHAC in Figure 12, where we picked six different policies during the training for the *Humanoid* problem, and plot the gradient value distribution of BPTT method and SHAC method for each of them. From Figure 12, we can clearly see that the gradient explosion problem exists in the BPTT method while the gradient acquired by SHAC is much more stable throughout the training process.

To better visualize the landscape, we also compare the landscape surfaces in Figure 13. Specifically, we experiment on the landscapes for the *Humanoid* and *Ant* problems. For *Humanoid*, we select a policy during training, and for *Ant* we use the optimal policy from training for study. We randomly select two directions $\vec{\theta_1}$ and $\vec{\theta_2}$ in the policy network parameter space, and evaluate the policy variations along these two directions with weights $\theta' = \theta + k_1\vec{\theta_1} + k_2\vec{\theta_2}, k_1, k_2 \in [-1, 1]$. The figure shows that the short-horizon episodes and the terminal value critic of SHAC produce a surrogate landscape which is an approximation of the original landscape but is much smoother.

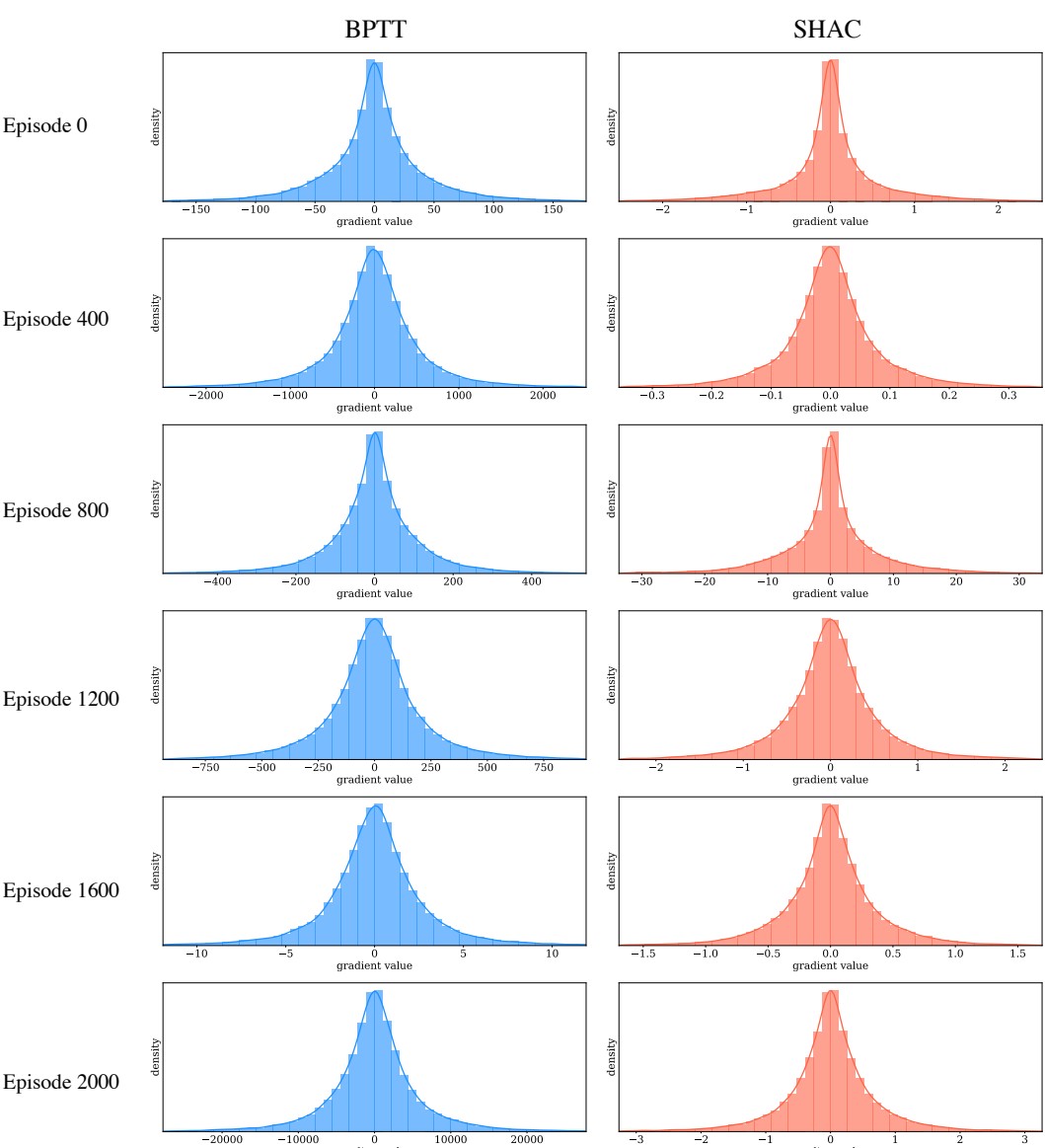

Figure 12: **Gradient distribution evolving over training process.** We select six policies from different episodes during the training process for the *Humanoid* problem and plot the gradient value distribution of BPTT (left) and SHAC (right). Note the $x$-axis scales on the plots. The plots show that the gradient computed from BPTT suffers from gradient explosion, whereas the analytical gradient computed from our method is much more stable.

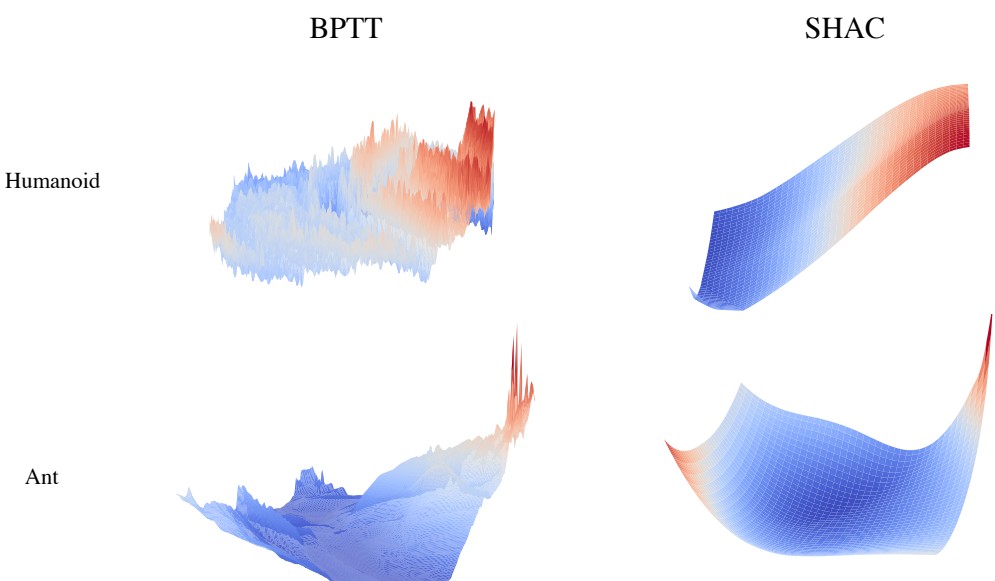

BPTT        SHAC

Humanoid

Ant

Figure 13: **Landscape surfaces comparison between BPTT and SHAC.** We study the landscapes for the *Humanoid* and *Ant* problems. For *Humanoid*, we choose a policy during the optimization and for *Ant* we choose the best policy from the learning algorithm. For each problem, we randomly select two directions in the policy parameter space and evaluate the policy variations along these two directions. The task horizon is $H = 1000$ for BPTT, and the short-horizon episode length for our method is $h = 32$. As we can see, longer optimization horizons lead to noisy loss landscape that are difficult to optimize, and the landscape of our method approximates the real landscape but is much smoother.

