# OpenReview forum: "Accelerated Policy Learning with Parallel Differentiable Simulation"
_ICLR.cc/2022/Conference — ICLR 2022 Poster_

### Official Review · Reviewer_5Q2y · 2021-10-18

**Correctness:** 4
**Technical Novelty And Significance:** 3
**Empirical Novelty And Significance:** 3
**Recommendation:** 8
**Confidence:** 4

**Main Review:**

**Strengths**
- The experimental results are impressive, and show the utility of using differentiable models.
- The method is sensible and well-motivated.
- Clarity is generally good, and method is explained in sufficient detail.
- The experimental procedures seemed well done, and they included ablation studies on the necessity of the critic and the used short horizon length.

**Weaknesses**
- Novelty is somewhat low.
- Discussion of gradient explosion seems inaccurate.
- Some discussion of related work is missing, particularly about related works in the model-based RL literature.
- Sometimes they are stating things as facts without providing evidence.
- There was no detailed discussion of how the results with the current simulator relate to other recent simulators such as Brax. Are the wallclock times similar? The results seem mixed, but having a detailed discussion would have been useful.
- The simulator is fast, so it may have been better to do more than 4 environments.

**Recommendation**

I recommend accepting the paper as I have not seen differentiable simulators used for tasks of the difficulty considered here (while also
taking advantage of differentiating through the simulator). I was considering a score of 6 or 8, but gave 6 for now.

**Discussion of points brought up**

Novelty:
It seems the contribution is primarily one of engineering, and they don't propose any surprising new idea. The idea of truncated backpropagation is old. Moreover, the policy training scheme resembles that of the the Dreamer algorithm (Hafner et al. 2019). Dreamer also uses short horizon rollouts together with a terminal value function, and backpropagates through these short horizons to optimize the policy. The differences are only: Dreamer uses lambda return weighting during the short horizons (why didn't you use this?), and the method of constructing the sequences is different (Dreamer samples start states from a replay buffer and performs model rollouts from these states, while the current work splits an episode into chunks). I am surprised that Dreamer was not discussed when explaining the methodology. Certainly it should be mentioned that there are prior works using a similar policy optimization procedure (with slight variations).

Discussion of gradient explosion:
There are other earlier works that do a more detailed job of discussing the gradient and landscape issues, such as PIPPS (Parmas et al. 2018), which should have been cited and discussed (moreover your methodology was very similar to these previous works). In your work, if gamma were 1, the value function were perfect and the policy were deterministic, the gradient you compute with your method should be exactly the same as the gradient that is computed using BPTT. From this point of view, your discussion is insufficient, as you do not explain why the loss landscape and gradients end up being smooth despite this fact. I can think of two possible reasons: 1. The value function is an approximator that ends up being smooth because of limited capacity to model the complicated landscape. 2. You are using a stochastic policy, and this stochasticty smooths out the landscape that the value function is estimating; hence it becomes smooth. However, one of the points brought up by Parmas et al. (2018) was that even if the landscape is smooth (due to averaging over policy or model stochasticity) the gradients computed by backpropagtion can be ill-behaved and lead to an explosion of the gradient variance.

For the other points see the raw notes at the bottom of this section.

Parmas, P., Rasmussen, C. E., Peters, J., & Doya, K. (2018,
July). PIPPS: Flexible model-based policy search robust to the curse
of chaos. In International Conference on Machine Learning
(pp. 4065-4074). PMLR.

**Questions**

Q1. In section 3.1 you write that you built the simulator. But from section A.3.1. it seems you just used Isaac Gym. So which one is it: did you make the simulator or did you use Isaac Gym?

Q2. In Figure 2, did you use a deterministic or stochastic policy? Was this the same policy as was used during training the value function? If the policy was stochastic, then how did you evaluate the landscape? This would require sampling many trajectories with the same policy and averaging. Are the scales on the left and right figures the same?

Q3. Did you do any ablation study of the policy noise? Does the method still work when the policy is deterministic? How much does the performance drop?

**Additional notes made during reviewing**

"for systems ranging from robots (e.g., Cheetah, Shadow Hand) to
complex anima- tion characters (e.g., muscle-actuated humanoids) using
only high-level reward definitions."
Please provide references.

"A differentiable simulator provides accurate first-order gradients of
the task performance reward with respect to the control inputs."
This is speculation. You provide no evidence. Problems with accuracy
can arise when the task performance depends on a sampled initial
state (so that the gradients are inherently stochastic).
Perhaps change to "may provide".

"However, despite the availability of differentiable simulators, it
has not yet been convincingly demonstrated that they can effectively
accelerate policy learning in complex high-dementional and
contact-rich tasks, such as some traditional RL benchmarks."
While this may be the case for differentiable simulators, there are
several model-based RL works that showed effective learning (e.g.,
Dreamer. The difference with a simulator is just that
the model does not have to be learned. I think the claim here is
downplaying such previous contributions. Also, "dementional" should be
"dimensional".

"There are several reasons for this: 1.(), 2.(), 3.()"
These reasons are stated as facts, while they are speculations.
Perhaps, "possible reasons" would be better. At least points 1 and 3
should be the same for model-free RL, so are they really the reason?
No references were provided.

"Because of these challenges, previous work has been limited..."
How do you know that those were the challenges that limited the
applicability of the previous methods? The publications themselves
do not seem to note your reasons as the reason why they limited their
experiments. For example, the PODS paper says their method overcomes
the issues of exploding gradients.

"In addition, we propose a truncated learning window to shorten the
backpropagation path to address problems with vanishing/exploding
gradients and reduce mem- ory requirements."
This is known as "truncated backpropagation". It is plagiarism to
claim that you "proposed" this.

In Equation 1, please provide the definition of the Jacobian's and
gradients. Usually, the gradient is a row vector, whereas you are
using a column vector.

"This makes the reward function smoother..."
What do you mean by this? The reward function is the same in all cases.

"In addition, we apply state normalization as is common in RL
algorithms."
Please explain what "state normalization" is.

"First, the terminal value function absorbs the discontinuity of long
dynamics horizons and early termination into a smooth function, as
shown in Figure 2 (Right)."
This explanation is incomplete. If you had no discount factor,
a perfect value model and a deterministic policy, your computed
policy gradient would be exactly the same as that of BPTT.

"Finally, the use of short horizons allows us to update the actor more
frequently, which, when combined with parallel differentiable
simulation, results in a significant speed up of training time."
Do you have an ablation study showing that it speeds up?

"In contrast, our method scales well due to direct access to the true
gradients from differentiable simulation."
You don't have access to true gradients when you are using a
stochastic policy. It may be better to explain this by referring to
the fact that reparamterization gradients are often more accurate
when computing gradients of smooth functions.


**Summary Of The Paper:**

They implement a differentiable simulator based on PyTorch (actually it was a bit unclear to me whether they implemented it, or just used Isaac Gym; I have a question about this below, and hope to receive an answer from the authors), as well as a method to utilize the differentiable simulation to optimize the policy more efficiently. The method operates as follows:
1. It computes a batch of episodes each of length H using a stochastic policy.
2. It splits the episodes into subsequences of length h.
3. It adds a terminal value function to each sequence, as well as discount factors on the rewards (starting from no discounting at the beginning of each subsequence), and computes policy gradients via backpropagation for optimization.
4. The value function is learned by computing lambda return targets (note that lambda returns are not used in the policy optimization though).

There is also some discussion about exploding gradients that hinder optimization through long horizons, which is why they truncated their episodes into subsequences.

They perform experiments on cartpole swingup, ant, humanoid and humanoid MTU (a humanoid actuated by 152 muscle tendon units), and compare to PPO, SAC, Backpropagation through time and PODS. The proposed method (SHAC) outperformed the other methods both in terms of number of steps (by a large margin) as well as computation speed (though PPO was faster in terms of wall-clock time in the early stage of learning after which SHAC overtook it). Particularly on the humanoid MTU task, the final performance was much better than the other methods. In terms of wall-clock time, the required computation time was a few minutes for cartpole and ant, and around 1h for the humanoid tasks.

They also performed ablation studies looking into the necessity of the terminal value function (it was necessary) and also the length of the subsequences (intermediate length sequences were optimal).

Recently some similar simulators, e.g. Brax or Isaac Gym have been made, but these works have not yet shown impressive results by taking advantage of differentiating through the model.

**Summary Of The Review:**

The main convincing result from the paper are the empirical results showing that using a differentiable model speeds up learning by a large margin in terms of step count on challenging tasks. I have not seen such results using differentiable simulators before, so I think it warrants publishing. The work itself seems primarily an engineering contribution: the methodology and ideas are not particularly novel, but executed in a sensible way. The quality of the work was generally good. Some more related work could be discussed, and the writing could be improved to be more precise.

______________________________________________________________
**Update**

I have increased my score to 8, novelty & significance score to 3, correctness to 4, and confidence to 4.

The authors have done a good job of addressing the reviewer concerns, and have made several changes to the manuscript, including adding more references; running more ablation experiments, e.g. experiments with a deterministic policy; added 2 new environments, etc.

I also note that their approach by differentiating through the trajectory and adding a terminal value function is not exactly the same as previous methods, such as Dreamer or MAAC. These previous works sample start states from a replay buffer of real data, and perform short rollouts from these states. The current work, on the other hand, operates on-policy, and it samples the start states from the end of the previous short rollout (this process is repeated until the total length of the short rollouts exceeds the episode length, and then the episode is restarted).  The policy is updated after each rollout, resembling a more typical truncated backpropagation procedure, but while adding a terminal value function. I think such an approach is tailored to using a simulator, because with a model-based approach, the predictions may diverge from the real trajectory when the task horizon is too long. On the other hand, if using a replay buffer based approach, such as is used in Dreamer or MAAC, the simulations to generate the replay buffer may be a waste of computational resources, so the approach presented in this paper seems like a natural idea to use in the setting when one has a simulator, and is not a direct copy of the previous works. The authors have also added a discussion of previous model-based RL methods compared to their work into Appendix A.4.6.

---

> ### Author Response · Authors · 2021-11-15
> **Response to Reviewer 5Q2y (1)**
>
> We sincerely thank reviewer 5Q2y for the feedback on our paper. We address the reviewer’s concerns as follows:
>
> &nbsp;
>
> **Q: Is the simulator just Isaac Gym?**
>
> **A:** Our simulator shares some ideas as Isaac Gym where both simulators accelerate the simulation via massive GPU parallelization. We have expanded this idea through our own end-to-end differentiable simulator written in PyTorch. We intend to open-source this framework to serve as a tool for future work.
>
> &nbsp;
>
> **Q: How is Fig 2 plotted?**
>
> **A:** When plotting Fig 2, we take a stochastic policy from the training (for *Humanoid* problem), and change one single weight of the policy within a range [-1, 1] to generate different policy variants. We fix the random seed before each policy variant evaluation to guarantee the initial state randomization and policy inference are consistent across different policy variants. We evaluate each policy variation with 128 trajectories and take the average loss. The scales on the left and right in Fig 2 are not exactly the same due to the imperfectness of the value function (the y axis in left plot is [-672, -629], while the y axis in right plot is [-388, -376]).
>
> &nbsp;
>
> **Q: Does the method still work when the policy is deterministic?**
>
> **A:** Our method does not constrain the policy to be deterministic or stochastic. As suggested, we have additionally run our method on all problems with **deterministic policy** without changing other hyperparameters. We ran on each problem with five random seeds and added the training plot in Appendix A.5.4 (Fig 9) in the updated paper. From the new plot, we can see the performance of deterministic policy is quite close to (sometimes even better than) the performance of the stochastic policy in the *CartPole Swing Up, Humanoid,* and *Humanoid MTU* problems. On the *Ant* problem, one random seed (out of five) fails to get a good policy resulting in a slight performance drop in this problem.
>
> &nbsp;
>
> **Q: Novelty is somewhat low.**
>
> **A:** Since this is a common question raised by multiple reviewers, we make two common replies in the **General Response** regarding the contribution of our work and a comparison between our method and the previous model-based RL methods.
>
> &nbsp;
>
> **Q: Lambda return weighting in the short horizon.**
>
> **A:** We are also using the lambda return for target value computation as shown in Eq (6).
>
> &nbsp;
>
> **Q: Discussion of gradient explosion seems inaccurate. If gamma = 1 and value function are perfect and the policy is deterministic, the gradient computed should be exactly BPTT’s gradient, why smoother? The smooth landscape in some cases can still result in an ill-behaved gradient when it is computed by averaging over policy stochasticity.**
>
> **A:** We would like to first mention that both plots in Fig 2 use the same gamma < 1. The main reason for the smoothed landscape is that the value function has its limited capacity and is parameterized to be a continuous function (i.e. each operator used in a network is a continuous function). Such a value function can serve as a continuous and smoother surrogate function of the real complicated and discontinuous landscape when the horizon is long. We also thank the reviewer for introducing the related papers ([PIPPS](https://arxiv.org/pdf/1902.01240.pdf)), which introduces a strategy to mix the log likelihood ratio gradient and the reparameterization gradient to obtain better gradient quality for stochastic trajectories. In our case, we only use the reparameterization gradient and use the short horizon reward plus a terminal value function to alleviate the landscape and gradient problems. The empirical results show that the reparameterization method works well in all of our problems. We also added a new plot in Fig 11 in Appendix A.6 about the comparison between the averaged analytical gradient and the finite difference gradient to show that the gradients computed by averaging over backpropagations is well-behaved. We agree that the mixed strategy proposed by PIPPS is a super interesting direction to explore in the future to further improve the gradient quality in the policy optimization.

---

> > ### Author Response · Authors · 2021-11-15
> > **Response to Reviewer 5Q2y (2)**
> >
> >
> >
> > **Q: How about the wallclock time compared to Brax?**
> >
> > **A:** Brax is a new simulation framework that appears to offer high-performance simulation of articulated dynamics. In our tests on the *Ant* problem, Brax is less than $1.5\times$ faster. While it is an interesting development we do not believe a direct comparison is meaningful for the following reasons.
> >
> > 1. We build our simulator on PyTorch which allows us to leverage many existing high-performance RL implementations (mentioned by Reviewer [nSwn](https://openreview.net/forum?id=ZSKRQMvttc&noteId=3IzY97_Ggc)).
> > 2. No benchmarks of Brax in reverse-mode (back-propagation) have been published.
> > 3. The Brax physical model is based on penalty models of joint constraints. While this is efficient to compute, it introduces significant error (displacement) at joints. In contrast, our simulator uses an industry-standard reduced coordinate model of joints that, while more expensive, ensures zero-joint error through direct factorizations.
> >
> > &nbsp;
> >
> > **Q: More environments.**
> >
> > **A:** We appreciate the reviewer’s acknowledgement on the potential bigger contribution of the developed differentiable simulation and the associated benchmark problems. As suggested to incorporate more environments, we added one more classical RL environment *HalfCheetah,* as suggested by Reviewer [nSwn](https://openreview.net/forum?id=ZSKRQMvttc&noteId=3IzY97_Ggc), and plan to add another one, *Hopper*, in the following week. The comparison results on the new problem is provided in Figure 7 in Appendix A.5.1. Furthermore, we will publish the parallel differentiable simulator as an open-source library upon acceptance, and keep the maintenance of the codebase to incorporate more environments in the future.
> >
> > &nbsp;
> >
> > **Q: Suggestions on writing/clarity.**
> >
> > **A:** We have made the suggested expositional changes, added suggested references, changed the gradient to be a row vector, and added a discussion about model-based RL in the related work section.
> >
> > &nbsp;
> >
> > **Q: The three reasons listed for previous differentiable simulation based methods are speculations.**
> >
> > **A:** We summarize these problems based on previous experience with differentiable simulations. We agree that the first point (local minimum problem) exists in model-free RL as well. However, by using the policy gradient estimated from Monte-Carlo samples, model-free RL has extra stochasticity and a non-local gradient estimation to help get out of local optima better than the methods based purely on the analytical local gradients. The third point (discontinuous landscape introduced by early termination) is not a big problem for model-free RL. The policy gradient acquired in model-free RL is still useful around discontinuous regions since it estimates the gradient from the zero-order information of Monte-Carlo samples. In contrast, in a pure analytical local gradient-based method (as used in most diffsim papers), the gradient information around the discontinuous region can be hard to use since the local gradient only reflects the function space on a single side of the discontinuity point. For example, a policy originally being able to survive the robot for 10 time steps will not get any information from the local gradient that it can actually obtain higher reward by surviving longer. Instead, the local gradient only provides the direction to a better policy within the first 10 time steps which may result in a suboptimal solution.
> >
> > &nbsp;
> >
> > **Q: Claim about "the use of short horizons allows us to update the actor more frequently, which results in a significant speedup of training time".**
> >
> > **A:** We support this claim using the ablation study of the short horizon length in the experiment section, as shown in Fig. 6. Intuitively, since the speed of each learning episode is determined by the length of a single trajectory, shorter trajectories result in more frequent updates. For example, simulating 100 trajectories of length 32 spends similar time to simulating 50 trajectories of the same length since the trajectories are simulated in parallel. However, simulating 50 trajectories of length 64 will take twice the time (even though the number of total samples collected rollouts samples are the same, e.g.: 100 * 32 = 50 * 64 = 3200 steps). Therefore, we can apply gradient-based updates more frequently if we use shorter horizons for the trajectories.

---

> > ### Comment · Reviewer_5Q2y · 2021-11-16
> > **A few clarifications and further comments**
> >
> > Thanks for the response, I'm especially happy to see the ablation study comparing with deterministic policies, and the extended discussion on the loss landscapes.
> > I have a few more comments on the current draft, and some clarifications about my previous review.
> >
> > First, one misunderstanding I had was that I thought that the episode is computed in full before any gradient updates are applied. But actually, after each short horizon, you compute the gradient, update the policy, and then continue computing the episode from where you left off. I find this interesting, but I would suggest to further clarify/emphasize it in the paper. For example, what you are referring to as a short horizon "episode" differs from what is usually considered an episode, so I would suggest to change the terminology that you are using.
> >
> > Regarding the additional results you have added in the appendix in A.5, perhaps you are planning to still revise the paper, but the additional results should also be referred to somewhere in the main paper. For example, the deterministic policy ablation could be referred to on page 4, when you mention deterministic/stochastic policies (or some other place in the paper). Also you are not the first to perform similar loss landscape analyses, so you should cite related work in section 3.2 and refer to the appendix for more details.
> >
> > Regarding lambda return weighting, I also wrote in my review that you are using this to compute the value targets. However, my comment was about using lambda return weighting also when computing the policy gradient, similarly to what Dreamer does. I wouldn't say it's a necessary comparison, but it would be interesting to know whether this may further improve the performance.

---

> > > ### Author Response · Authors · 2021-11-17
> > > **Response to new comments**
> > >
> > > Thanks for providing more insightful suggestions. We address the reviewer’s suggestions as follows:
> > >
> > > &nbsp;
> > >
> > > **Q: Clarify the short horizon “episode” in the paper.**
> > >
> > > **A:** We apologize for the confusion, and agree the term short-horizon is somewhat overloaded. As suggested, we have made the following changes in the revised manuscript to clarify the algorithm:
> > >
> > > 1. We use the term `task horizon` for the original horizon of the control task to help remind the reader.
> > > 2. We use the term `learning episode` to indicate it to be the episode of the learning algorithm instead of the episode of the trajectory/task.
> > > 3. We rephrased some text in Section 3.3, Figure 3 and Algorithm 1 to clarify that the algorithm splits the whole long trajectory into short sub-trajectories across learning episodes.
> > >
> > > &nbsp;
> > >
> > > **Q: Refer to the additional results in the main paper.**
> > >
> > > **A:** Thank you for this valuable suggestion. We have made the following changes:
> > >
> > > 1. In section 3.2, we add a reference to PIPPS and point the readers to the loss landscape analysis in the Appendix.
> > > 2. In section 4.1, we add text to refer the readers to the deterministic policy experiments and the fixed hyperparameter experiments in the Appendix.
> > > 3. In section 4.2, we add text to refer the readers to the newly implemented environments in the Appendix.
> > > 4. In section 4.3, we add text to refer the readers to the new comparison experiments with REDQ in the Appendix.
> > >
> > > &nbsp;
> > >
> > > **Q: Use lambda return weighting for policy gradient computation.**
> > >
> > > **A:** Apologies for our misunderstanding of the original question. This is a very good point. We actually tried to use the lambda return weighting for the policy loss computation, however the early-stage results showed that it did not produce better results than the current method and thus we did not adopt it. However, we agree this is an interesting direction to explore further in the future.
> > >
> > > ---
> > >
> > > In addition, we add the second new environment *Hopper* and the corresponding experiment plot in Appendix A.3.3 and Appendix A.4.1

---

> > > > ### Comment · Reviewer_5Q2y · 2021-11-24
> > > > **Updated review**
> > > >
> > > > Thanks, I have updated my review.

---

### Official Review · Reviewer_nSwn · 2021-11-02

**Correctness:** 4
**Technical Novelty And Significance:** 2
**Empirical Novelty And Significance:** 4
**Recommendation:** 8
**Confidence:** 4

**Main Review:**

Pros:
* The paper is clearly written and well organized.
* The differentiable simulator introduced will likely have great value to researchers, as not only does it have the ability to be differentiable, but being built on PyTorch means that it is likely more interpretable and customizable than similar methods [1], and is also natively parallelizable, which will result in massively reduced experimentation times.
* The authors show very convincing performance of their introduced SHAC method on this suite of benchmarks compared to competitive methods, including non-analytical policy gradient approaches (PPO/SAC), as well as other methods that leverage differentiable environment dynamics.

Cons:
* Key design elements and motivations in the SHAC method have appeared in different guises in the literature before, and some of these aren't referenced. For instance, in MAAC [2], they too have an analytical policy gradient (APG) (due to differentiability of the world model), and then prevent issues of long-horizon BPTT by including a truncating value function. Going further back, in [3] the idea of terminating rollouts using a value function is also used/introduced (in their case also to improve computation time). Indeed, the paper cited by the authors [4] which uses 'enhancement' can be viewed as a one-step analytical policy gradient with termination. Perhaps the authors could describe why having a state-value terminal function (instead of an action-value function which supports action gradients) is beneficial in their setting? Also I am somewhat concerned by the use of different architectures for each task; could the authors run some sensitivity analysis by keeping architecture fixed for each task to demonstrate the robustness of SHAC to this?
* Similarly, given the relative simplicity of the SHAC method, it's somewhat disappointing that the authors only introduced 4 environments, and no manipulator-based contact environments (which would surely be a good fit for the goal of smoothened contact physics). In my opinion, focus should have been placed more on either the method or the simulator, but at this point for me neither display the requisite gravitas. I would like to see some additional environments introduced, such as a Cheetah and Hopper derivatives, and this would help the adoption of such an environment to a wider RL audience.
* Are the smoothness assumptions made in the simulator actually transferable to real life robotics? There should be discussion as to whether or not the modelling choices made are realistic (e.g., smooth joint limits, choice of differentiable contact physics). As has been pointed out in the past, non-APG methods can be successfully deployed on real robots without requiring differentiability [5], whereas APG approaches will need to to get over the issue of `sim2real`. I can see this in fact being further exacerbated by the differentiability of the simulator, which would allow the policy to very effectively exploit the simulator to obtain great solutions that don't transfer (for instance, see [6] where the solutions are able to exploit deficiencies the MuJoCo simulator). In this case, perhaps the humanoid MTU solutions of SAC aren't as optimal in the simulator, but could transfer better IRL. Is there a way the authors could show that this exploitation is avoided in their simulator?
* I found the single weight perturbation method to demonstrate smoothness slightly odd as there is abundant literature that aims to visualize loss surfaces, such as [7]. Why did the authors elect to use their single weight perturbation approach?

Refs:

[1] Brax -- A Differentiable Physics Engine for Large Scale Rigid Body Simulation: Freeman et al., arXiv:2106.13281

[2] Model-Augmented Actor-Critic: Backpropagating through Paths: Clavera et al., arXiv:2005.08068

[3] Value Function Approximation and Model Predictive Control: Zhong et al., ADPRL ‘13

[4] Efficient Differentiable Simulation of Articulated Bodies: Qiao et al., arXiv:2109.07719

[5] Soft Actor-Critic Algorithms and Applications: Haarnoja et al., arXiv:1812.05905

[6] On the Importance of Hyperparameter Optimization for Model-based Reinforcement Learning: Zhang et al., arXiv:2102.13651

[7] Visualizing the Loss Landscape of Neural Nets: Li et al., arXiv:1712.09913


**Summary Of The Paper:**

In this paper, the authors introduce a novel differentiable simulator (built on PyTorch) that focuses on enabling smooth gradients with respect to expected return, especially with a view to ameliorating discontinuities in contact dynamics. In order to leverage this differentiable engine as effectively as possible, they also introduce SHAC, which is an analytical policy gradient (APG) method that features a terminal value function to truncate the value gradient calculation through time. The authors' show this results in a smoother objective function with lower training time, which empirically gives rise to faster/improved convergence in policy learning.

**Summary Of The Review:**

Overall the paper introduces a novel differentiable environment suite, and aims to ameliorate a key issue in analytical policy gradients. However I believe there are deficiencies in both these things that I have listed above, so recommend a weak rejection for now.

Having said this I believe this paper has great promise, and think these issues are not major, and am happy to raise to an accept if they are sufficiently addressed.

--------------- POST REBUTTAL ------------------

The authors have addressed most of my concerns, including the sensitivity of their approach to architecture, and included new environments aligned with canonical continuous control problems. Furthermore they focused more on the simulator in their updates, which is where I believe most of the impact of this paper lies, and produced additional interesting analysis about loss landscape smoothness.

---

> ### Author Response · Authors · 2021-11-15
> **Response to Reviewer nSwn (1)**
>
> We sincerely thank reviewer nSwn for the feedback on our paper. We address the reviewer’s concerns as follows:
>
> &nbsp;
>
> **Q: Lack of technical novelty.**
>
> **A:** Since this is a common question raised by multiple reviewers, we make two common replies in the **General Response** regarding the contribution of our work, and a comparison between our method and the previous model-based RL methods.
>
> &nbsp;
>
> **Q: Comparison with the "**[policy enhancement](http://proceedings.mlr.press/v139/qiao21a/qiao21a.pdf)**" approach. Why having a state-value terminal function?**
>
> **A:** Our choice of state value function is just a design choice of our method since it is common to use a state value function instead of state-action Q function in on-policy policy gradient methods such as PPO. The difference between these two choices is simply whether we want to explicitly include the action in the value function and use the action gradient on the terminal step. When compared to the "policy enhancement" approach proposed by [Qiao et. al.](http://proceedings.mlr.press/v139/qiao21a/qiao21a.pdf), the main differences are:
>
> 1. Our method uses of a longer horizon length which allows us to exploit the benefit of the simulator more (as shown in the Ablation study in Fig 6);
>
> 2. While our method optimizes the policy in an on-policy mode, Qiao et. al. integrates their "enhancement" into a model-based algorithm with off-policy updates (MBPO), compared to which, our method has much faster training time and higher achievable rewards;
>
> 3. The "policy enhancement" method has been shown by Qiao et. al. to be not effective on problems other than pendulum. In contrast, our method is demonstrated to work successfully on different tasks spanning a wide range of problem complexities.
>
> &nbsp;
>
> **Q: Concerns about using different architectures for each task. Some sensitivity analysis by keeping architecture fixed for each task is helpful.**
>
> **A:** Thank you for this valuable suggestion to help improve the paper. We have added the suggested sensitivity analysis by using the same architectures and hyperparameters (e.g. learning rates) as used in the *Humanoid* problem for all four problems, and added the comparison plots in the Appendix A.5.3 (Fig 8). The results show that our method still works reasonably well with this fixed hyperparameter choice.
>
> In the main manuscript, we select different architectures and learning rates for different tasks because:
>
> 1. It is natural to use larger networks for more complex problems with larger observation/action spaces while using relatively smaller networks for lower-dimensional problems.
> 2. We tuned different architectures and hyperparameters for different tasks in order to show the maximal performance (e.g. highest reward, fastest running time, best sample efficiency) of each algorithm on each problem (we tuned for both ours and all baseline algorithms) for a completely fair comparison in their optimal settings. This is also a practice adopted by a recent benchmark work ([Isaac Gym](https://arxiv.org/pdf/2108.10470.pdf)).
>
> &nbsp;
>
> **Q: More environments**
>
> **A:** We appreciate the reviewer’s acknowledgement on the potentially bigger contribution of the developed differentiable simulation and the associated benchmark problems. As suggested to incorporate more environments, we have added two more classical RL environments *HalfCheetah* and *Hopper*. The comparison results on the new problems are provided in Figure 7 in Appendix A.5.1. Furthermore, we will publish the parallel differentiable simulator as an open-source library upon acceptance, and keep the maintenance of the codebase to incorporate more environments such as manipulation tasks in the future.

---

> > ### Author Response · Authors · 2021-11-15
> > **Response to Reviewer nSwn (2)**
> >
> > **Q: Are the smoothness assumptions made in the simulator transferable to real life robotics?**
> >
> > **A:** Thanks for raising this interesting question. There are two main applications that can benefit from our method, **physics-based animation** and **robotic control**. For physics-based animation (e.g. the *Humanoid MTU* example), our method can significantly accelerate the policy optimization process and small modifications on the dynamics models are usually allowed. For robotics, the sim-to-real problem is more important. While a full treatment of the sim-to-real problem is outside the scope of this work, we believe our method is not fundamentally limited in this aspect for the following reasons:
> >
> > 1. The smoothed contact models and joint limits we used in the simulation dynamics are inspired from two previous differentiable simulation papers, namely [ADD](https://arxiv.org/pdf/2007.00987.pdf) and [Diff Redmax](http://www.roboticsproceedings.org/rss17/p008.pdf). Both papers show real robot examples which demonstrate that the adopted physics model has its potential matching to real-world physics.
> > 2. There has been some great work in RL [1, 2, 3] showing the possibility of sim-to-real transfer via using stochastic policies and applying some techniques such as domain randomization during training to improve the robustness of the learned policy. Such techniques can also be easily applied in our method for sim-to-real transfer.
> > 3. Differentiable simulators actually open up new possibilities for sim-to-real transfer via accelerating system identification with the access to the simulation parameters’ gradients [4]. Through system identification, we can calibrate the simulation to match better to the real world, which had to be done with a gradient-free method previously without the access to a differentiable simulator.
> >
> > Given this, we believe that addressing the sim-to-real problem for differentiable simulation is an exciting and promising direction to be explored in the future.
> >
> > &nbsp;
> >
> > **Q: Single weight perturbation for Fig 2 is weird, why not visualize loss surface?**
> >
> > **A:** The single weight perturbation used in Fig 2 can be regarded as a variation of the [1D linear interpolation method](https://arxiv.org/pdf/1412.6544.pdf), which has also been commonly used in [previous literature](https://arxiv.org/pdf/1902.01240.pdf) despite some problems, as pointed out by the [paper](https://arxiv.org/abs/1712.09913) the reviewer mentioned. Our goal of plotting the landscape is to give an intuitive and easy-to-understand illustration of how noisy the original landscape is and how the surrogate landscape looks like after our improvement. We found that the single weight perturbation method was sufficient for us to achieve this goal. We have further expanded on this point in the revised manuscript by including a plot of gradient magnitude distributions in Fig 12 in Appendix A.6 (as suggested by Reviewer [4PEt](https://openreview.net/forum?id=ZSKRQMvttc&noteId=tMN_P29pV-)).
> >
> > ---
> >
> > [1] OpenAI : Marcin Andrychowicz, Bowen Baker, Maciek Chociej, Rafal Jozefowicz, Bob McGrew, ´Jakub Pachocki, Arthur Petron, Matthias Plappert, Glenn Powell, Alex Ray, Jonas Schneider, Szymon Sidor, Josh Tobin, Peter Welinder, Lilian Weng, and Wojciech Zaremba. Learning dexterous in-hand manipulation. The International Journal of Robotics Research, 39(1):3–20, 2020.
> >
> > [2] Jie Tan, Tingnan Zhang, Erwin Coumans, Atil Iscen, Yunfei Bai, Danijar Hafner, Steven Bohez, Vincent Vanhoucke. Sim-to-Real: Learning Agile Locomotion For Quadruped Robots. Proceedings of Robotics: Science and Systems, 2018
> >
> > [3] Yevgen Chebotar, Ankur Handa, Viktor Makoviychuk, Miles Macklin, Jan Issac, Nathan Ratliff, Dieter Fox. Closing the Sim-to-Real Loop: Adapting Simulation Randomization with Real World Experience. International Conference on Robotics and Automation (ICRA), 2019
> >
> > [4] Underwater Soft Robot Modeling and Control with Differentiable Simulation. Tao Du, Josie Hughes, Sebastien Wah, Wojciech Matusik, Daniela Rus. RA-L/RoboSoft 2021

---

> > > ### Comment · Reviewer_nSwn · 2021-11-18
> > > **Thank you for your response**
> > >
> > > My fundamental concern about novelty was still not really addressed, as I still believe the SHAC optimizer is very similar to prior work, as has been highlighted by other reviewers. It is true that minor design choice modifications are required to adjust from model-based methods into what is an on-policy model-free approach in a dynamics-smooth environment, but I didn't find these changes surprising nor interesting, which is where I believe the authors argue their novelty lies.
> > >
> > > I believe the main merit of this paper lies with the differentiable PyTorch simulator, and framing this in a more RL-friendly narrative (as has been done in this work) has great merit to the wider community. The further addition of new environments in the rebuttal has greatly improved this aspect of the paper, and therefore I am willing to raise my score. How I currently view this work is mainly as introducing a new problem setting/analytical tool (e.g., smooth, differentiable, parallelisable environments that are easy to parse and build on in PyTorch), and also provide a logical approach (i.e., SHAC) that leverages these properties. I also agree with the authors that there is future interesting work that should be explored (such as `sim2real`, the impact of differentiability in physics-based simulation), and their simulator provides a foundation from which to do this.
> > >
> > > Weighing up these things I am happy to raise my score, but am still not comfortable moving to an accept (i.e., 8) as I believe too much emphasis is on the policy optimizer as compared with the simulator, which has limited the opportunity to explore more meaningful research questions about the simulator that I believe would be more of interest to the wider RL community.
> > >
> > > P.S.: I am aware that their weight perturbation approach is a special case of the work I cited (e.g., what if the random direction sampled was along a single parameter). Therefore there is a very simple fix to align with the literature, which is to simply visualize the objective along line paths that change multiple parameters, achieved by sampling that direction randomly. I could imagine a 3D plot similar to introduction of [the paper I mentioned](https://arxiv.org/abs/1712.09913) could be a very powerful visual aid to demonstrate smoothness!

---

> > > > ### Author Response · Authors · 2021-11-21
> > > > **Response to new comments**
> > > >
> > > > Thanks for your reply and the valuable suggestions. To put more focus on the gpu-parallelized simulation, we have added substantial details of the simulation in Section 3.1 in the main paper, removed/shortened part of the SHAC algorithm description in Section 3.3, and highlighted the simulation in the abstract and introduction. As suggested, we have also added the landscape surface plots for Humanoid problem and Ant problem in Figure 13 in Appendix A.5.
> > > >
> > > > Furthermore, in terms of the novelty, we believe the contribution of our work consists of two parts.
> > > >
> > > > 1. First, our work contributes to the **RL community**, as the reviewer acknowledged, through a **parallel differentiable simulator**, which in tandem with policy gradient and actor-critic methods from RL significantly improves the efficiency of the policy learning. We have implemented a differentiable simulator that features a GPU-parallelization scheme and adopts a smooth contact/joint limit model. **As suggested, we have added substantial details in the Section 3.1 in the updated manuscript to emphasize its contribution**, and will publish the simulation framework openly to encourage future work that builds on this.
> > > >
> > > > 2. In addition, this work has a potentially strong contribution to the **differentiable simulation community**, which might be overlooked by the reviewer. While the recent interest in differentiable simulation opens up new possibilities, how to effectively use these new tools to solve sophisticated control tasks is still an open problem. For example, in Section 5.1.4 of the recent [Brax](https://arxiv.org/pdf/2106.13281.pdf) paper on analytic policy gradient methods:
> > > >
> > > >    > This algorithm is less mature than the previous three, and does not currently produce locomotive gaits, and instead seems prone to being trapped in local minima on the environments we provide. Differentiating through long trajectories is an active area of research and is known to be difficult to optimize, thus we defer more advanced differentiable algorithms to future releases.
> > > >
> > > >    **Our paper is a timely attempt to fill this gap** by first analyzing the reasons behind the optimization difficulties and then presenting an effective idea which combines model-based RL with analytical gradient computation from differentiable physics to **solve control problems at complexity levels that have never been achieved by differentiable simulation based methods before** (also credited by Reviewer [5Q2y](https://openreview.net/forum?id=ZSKRQMvttc&noteId=L72ig3zxfB) and [4PEt](https://openreview.net/forum?id=ZSKRQMvttc&noteId=tMN_P29pV-)). Linking these two communities together gives our work a unique value, and serves as a strong baseline for the future work using differentiable simulation.
> > > >
> > > > ---
> > > >
> > > > **Compared to MAAC**
> > > >
> > > > Notably, the argument is that MAAC is not the first in proposing this idea of gradient computation, since the idea originates in [Nguyen et al. 1990](https://ieeexplore.ieee.org/document/55119), and [Jordan et al. 1992](https://www.sciencedirect.com/science/article/abs/pii/036402139290036T). The difference between recent methods is often how the gradient is computed! MAAC, while impressive in results, could itself be treated as a variant of [SVG](https://arxiv.org/pdf/1510.09142.pdf) with a H-step rollout (denoted as SVG(H) in a recent work by [Amos et al. 2021](https://arxiv.org/pdf/2008.12775.pdf)). In a similar vein, SHAC extends these ideas to use a TD-lambda style value estimation rather than using value expansion with a fixed H (adopted by MAAC), adopt an on-policy update mode, and use analytically differentiable dynamics models to avoid model error which enables to use longer model rollout horizon (32 vs shorter than 10 in Fig 3 in [MAAC](https://arxiv.org/pdf/2005.08068.pdf)). All these modifications together enable better efficiency and scale-up to higher dimensional problems.

---

> > > > > ### Comment · Reviewer_nSwn · 2021-11-21
> > > > > **Thanks!**
> > > > >
> > > > > Thank you for the reply and updates. Looking at the manuscript now, I'm much happier with the contribution of this paper, and indeed believe it is now a valuable addition to the differentiable simulator literature. In short, most of my concerns have now been addressed and I'm happy to recommend accepting this paper. I also appreciate the authors running the optimization landscape visualizations and expanding that aspect of the paper, as it provides further empirical justification for the SHAC method.

---

### Official Review · Reviewer_Hcaa · 2021-11-02

**Correctness:** 4
**Technical Novelty And Significance:** 2
**Empirical Novelty And Significance:** 3
**Recommendation:** 6
**Confidence:** 4

**Main Review:**

Overall the paper introduces an interesting idea of leveraging simulator derivative information to achieve fast and sample efficient reinforcement learning. The idea is well presented and I found the empirical results very convincing. However I do find a few shortcomings that I hope the authors can address.


Pros


1. The empirical results of the proposed method are very strong. It can be clearly seen that the proposed method outperforms baselines significantly. I consider this as the main advantage of this paper, since the proposed method and the differentiable simulator could serve as good foundations for future work.


2. The PyTorch GPU based differentiable simulator that supports complex environments is another important contribution of the paper. If released, it could be impactful for the field as it enables researchers to obtain controllers for complex systems much faster.


3. The paper is well written. The overall structure of the paper is well organized and the presentation of the main contribution is easy to follow.


Cons


1. I’m not convinced about the novelty of the proposed method. The proposed method is mostly the same as the method introduced in [1], where the only difference seems to be that the proposed method uses exact gradient from the simulator instead of gradient of a learned dynamics model. Although the proposed method is not new, a major novelty of this paper could come from the creation of the differentiable simulator that supports complex dynamics models such as the Humanoid MTU used in this paper. However unfortunately the simulator implementation is not discussed in detail in this paper.


2. The sample efficiency of off policy reinforcement learning methods has also improved a lot in recent years, where some model free methods can even achieve sample efficiency close to that of model based methods ([2]). It would be important to compare to some of these methods, because this comparison would tell us how much improvement can we get from using the derivative information of the simulator.


Given these limitations, I cannot recommend acceptance of this paper at its current state. I highly encourage the authors to continue improving the paper by providing more discussion regarding the simulator and adding the comparison to more recent reinforcement learning methods.


References


[1] Clavera, Ignasi, Violet Fu, and Pieter Abbeel. "Model-augmented actor-critic: Backpropagating through paths." arXiv preprint arXiv:2005.08068 (2020).


[2] Chen, Xinyue, et al. "Randomized ensembled double q-learning: Learning fast without a model." arXiv preprint arXiv:2101.05982 (2021).


## Update after Authors' Response
The authors have updated the paper to include more description of the simulator and the missing references. Focusing more the the differentiable simulator, the manuscript now reflects the contribution of the work more accurately and therefore I believe it is a valuable addition of the community. Now I recommend acceptance of this paper.


**Summary Of The Paper:**

This paper focuses on the setting of reinforcement learning with differentiable simulation, where the authors assume access to a simulator which can compute the derivatives of the next state and reward with respect to the current state and action exactly. Under this setup, the authors propose an actor-critic style reinforcement learning algorithm that leverages the simulator derivative information to achieve better training sample efficiency and wall-clock time. Specifically, the authors expand the value function of the policy in h steps, and combine the sum of the simulator gradient of these h steps and the gradient of the learned value function at the h + 1 step to obtain the overall gradient for training the policy. The value function is then trained in the same way as in other actor-critic style methods.

The authors evaluate the proposed algorithm empirically on a few domains and the experiment results suggested that the proposed method achieves superior sample efficiency and wall-clock time compared to prior methods.



**Summary Of The Review:**

The paper presents an intuitive reinforcement method for leveraging derivative information from a differentiable simulator. The methods significantly outperforms prior methods in terms of both sample efficiency and wall-clock time. However, the proposed method is not novel as it is largely the same as that introduced in a prior work, and it is missing an important baseline comparison. Therefore I cannot recommend acceptance of the paper in its current state.

---

> ### Author Response · Authors · 2021-11-15
> **Response to Reviewer Hcaa**
>
> We sincerely thank reviewer Hcaa for the feedback on our paper. We address the reviewer’s concerns as follows:
>
> &nbsp;
>
> **Q: Lack of novelty**
>
> **A:** Since this is a common question raised by multiple reviewers, we make two common replies in the **General Response** regarding the contribution of our work, and a comparison between our method and the previous model-based RL methods.
>
> &nbsp;
>
> **Q: More simulation details**
>
> **A:** We appreciate the reviewer’s acknowledgement on the potential bigger contribution of our differentiable simulator and the associated benchmark problems. Due to the space limit and some shared techniques with previous differentiable simulators, we choose to keep it concise and compact in the main paper. As suggested, we will include more details of the simulation in the Appendix in the upcoming week. At the same time, as promised in the reproducibility section we will publish the parallel differentiable simulator and the benchmark environments as an open-source library upon acceptance.
>
> &nbsp;
>
> **Q: Comparison to more advanced model-free RL algorithms (**[**REDQ**](https://arxiv.org/pdf/2101.05982.pdf)**).**
>
> **A:** Thanks for this valuable suggestion. As suggested, we have conducted additional experiments with the [official code](https://github.com/watchernyu/REDQ) released by REDQ and added the comparison plot in the Appendix A.5.5 (Fig 10). From the experiments on *CartPole Swing Up* and *Ant* problems, REDQ demonstrates its high sample efficiency due to its off-policy update and randomized Q function ensembling strategy. However, we find that REDQ is slow in wall-clock comparisons (e.g. in *Ant* problem, REDQ takes 27 hours to achieve 5000 rewards while SHAC only takes 10 minutes to achieve the same reward on the same machine). Furthermore, the official implementation and the default hyperparameters of REDQ fails to train on our *Humanoid* and *Humanoid MTU* problems.
>
> Due to the setting of learning policies in a fast simulation, we care about not only the sample efficiency but also the wall-clock time efficiency. In this setting, on-policy methods are stronger baselines to compare with. Thus in our experiments, we compare to a state-of-the-art implementation of PPO by [RL Games](https://github.com/Denys88/rl_games) which also leverages the parallel simulation to achieve the best balance between sample efficiency and wall-clock time efficiency which is also adopted by a recent RL benchmark work ([Isaac Gym](https://arxiv.org/pdf/2108.10470.pdf)). For the off-policy methods, we use the SAC implementation also provided by RL Games.

---

> > ### Comment · Reviewer_Hcaa · 2021-11-17
> > **Re: Author Response**
> >
> > I'd like to thank the authors for the detailed response to the reviews. The extra experiment results have addressed my concerns about comparing to better off-policy RL methods. However, my concern about the novelty remains.
> >
> > While the authors acknowledge that proposed method shares similarity with some prior methods, it still seems to me that the proposed method is a direct application of MAAC, where the learned dynamics model is replaced by a differentiable simulator, which by itself is also not a novel concept. In other words, if I take the MAAC algorithm and replace its learned dynamics model component with an existing differentiable simulator such as Brax, I would end up with pretty much exactly this method. Therefore, it is hard convince me that the proposed SHAC method is really a new way of leveraging a differentiable simulator in RL. On the other hand, much of the contribution of the paper comes from the implementation of the PyTorch differentiable simulator. I believe that if the authors could revise the paper to focus more on the simulator instead of the SHAC method, the paper would better reflect the contributions of the work.

---

> > > ### Author Response · Authors · 2021-11-21
> > > **Response to new comments**
> > >
> > > Thank you for your reply. To put more focus on the gpu-parallelized simulation, we have added substantial details of the simulation in Section 3.1 in the main paper, removed/shortened part of the SHAC algorithm description in Section 3.3, and highlighted the simulation in the abstract and introduction.
> > >
> > > Furthermore, in terms of the novelty, we believe the contribution of our work consists of two parts.
> > >
> > > 1. First, our work contributes to the **RL community**, as the reviewer acknowledged, through a **parallel differentiable simulator**, which in tandem with policy gradient and actor-critic methods from RL significantly improves the efficiency of the policy learning. We have implemented a differentiable simulator that features a GPU-parallelization scheme and adopts a smooth contact/joint limit model. **As suggested, we have added substantial details in the Section 3.1 in the updated manuscript to emphasize its contribution**, and will publish the simulation framework openly to encourage future work that builds on this.
> > >
> > > 2. In addition, this work has a potentially strong contribution to the **differentiable simulation community**, which might be overlooked by the reviewer. While the recent interest in differentiable simulation opens up new possibilities, how to effectively use these new tools to solve sophisticated control tasks is still an open problem. For example, in Section 5.1.4 of the recent [Brax](https://arxiv.org/pdf/2106.13281.pdf) paper on analytic policy gradient methods:
> > >
> > >    > This algorithm is less mature than the previous three, and does not currently produce locomotive gaits, and instead seems prone to being trapped in local minima on the environments we provide. Differentiating through long trajectories is an active area of research and is known to be difficult to optimize, thus we defer more advanced differentiable algorithms to future releases.
> > >
> > >    **Our paper is a timely attempt to fill this gap** by first analyzing the reasons behind the optimization difficulties and then presenting an effective idea which combines model-based RL with analytical gradient computation from differentiable physics to **solve control problems at complexity levels that have never been achieved by differentiable simulation based methods before** (also credited by Reviewer [5Q2y](https://openreview.net/forum?id=ZSKRQMvttc&noteId=L72ig3zxfB) and [4PEt](https://openreview.net/forum?id=ZSKRQMvttc&noteId=tMN_P29pV-)). Linking these two communities together gives our work a unique value, and serves as a strong baseline for the future work using differentiable simulation.
> > >
> > > ---
> > >
> > > **Compared to MAAC**
> > >
> > >    Notably, the argument is that MAAC is not the first in proposing this idea of gradient computation, since the idea originates in [Nguyen et al. 1990](https://ieeexplore.ieee.org/document/55119), and [Jordan et al. 1992](https://www.sciencedirect.com/science/article/abs/pii/036402139290036T). The difference between recent methods is often how the gradient is computed! MAAC, while impressive in results, could itself be treated as a variant of [SVG](https://arxiv.org/pdf/1510.09142.pdf) with a H-step rollout (denoted as SVG(H) in a recent work by [Amos et al. 2021](https://arxiv.org/pdf/2008.12775.pdf)). In a similar vein, SHAC extends these ideas to use a TD-lambda style value estimation rather than using value expansion with a fixed H (adopted by MAAC), adopt an on-policy update mode, and use analytically differentiable dynamics models to avoid model error which enables to use longer model rollout horizon (32 vs shorter than 10 in Fig 3 in [MAAC](https://arxiv.org/pdf/2005.08068.pdf)). All these modifications together enable better efficiency and scale-up to higher dimensional problems.

---

> > > > ### Comment · Reviewer_Hcaa · 2021-11-22
> > > > **Re: Author Response**
> > > >
> > > > Thanks for the detailed comments and changing the manuscript! Now that the main paper includes more description to the PyTorch based differentiable simulator I think it reflects the contribution of the paper much more accurately. I'm happy to increase my rating of the paper and recommend acceptance.

---

### Official Review · Reviewer_4PEt · 2021-11-04

**Correctness:** 4
**Technical Novelty And Significance:** 2
**Empirical Novelty And Significance:** 4
**Recommendation:** 8
**Confidence:** 4

**Main Review:**

Strengths:
* Idea is conceptually simple and provides strong results on a spectrum of established benchmark environments as well on one very high dimensional task.
* Shows the potential of backward differentiation through time.

Weaknesses:
* Could be run on other benchmark environments, which might highlight more the limits of the proposed approach.
* The analysis of why and whether the loss landscape is smoother could be more elaborate. Additional figures like for example a histogram of the gradient distribution evolving over time would, in my opinion, help the reader to have better insight than Figure 2.

Questions:
* How important is the structure of the reward function for this approach?
* Have you found out on which kind of environments the proposed method fails?

Additional Remarks:
* Typo in the introduction: high-dementional
* I found the truncation of the curves in Figure 4 visually misleading at first.
* If section 3.2 would be more elaborate and the experiment would include more environments, this paper could be further improved.

**Summary Of The Paper:**

This paper demonstrates the effective use of differentiable simulators for significantly speeding up the training of policy gradient methods. The main idea is to backpropagate through the simulator only on shorter partitions of the trajectory, which seems to lead to a smoother loss landscape and better gradient signal. To avoid getting stuck in local minima, a critic is used to incorporate a terminal cost. As the implementation can be parallelized and the simulation run on the GPU, the experiments show drastic improvements not only in sample efficiency but also in run time.

**Summary Of The Review:**

This paper shows that differentiable simulators can be leveraged to significantly speed up training even on very challenging high-dimensional tasks, making it an interesting empirical and conceptual contribution.

---

> ### Author Response · Authors · 2021-11-15
> **Response to Reviewer 4PEt**
>
> We sincerely thank reviewer 4PEt for the feedback on our paper. We address the reviewer’s concerns as follows:
>
> &nbsp;
>
> **Q: Could the experiments include more environments? When the algorithm fails?**
>
> **A:** To incorporate more environments, we have added two additional classical RL environments, *HalfCheetah* and *Hopper*, as suggested by Reviewer [nSwn](https://openreview.net/forum?id=ZSKRQMvttc&noteId=3IzY97_Ggc). The comparison results of SHAC and the baseline methods on the new problems are provided in Figure 7 in Appendix A.4.1. We intend to maintain the simulator as an open environment and increase the diversity of tasks over time.
>
> For the failure cases, we notice that SHAC fails to find a good policy with very few random seeds. As an example, when running our method with deterministic policy (as requested by Reviewer [5Q2y](https://openreview.net/forum?id=ZSKRQMvttc&noteId=L72ig3zxfB)), there is one seed (out of five) for the *Ant* problem that results in a suboptimal policy. However in the majority of runs, our method works well across different seeds and problems.
>
> &nbsp;
>
> **Q: More analysis of why and whether the loss landscape is smoother could be more elaborate. Providing a histogram of the gradient distribution evolving over time would be helpful.**
>
> **A:** **Whether loss landscape is smoother?** We have elaborated in Section 3.2 about how noisy the original landscape of the problem is (corresponding to Fig 2 left), and discuss in the last paragraph of Section 3.3 about how our method SHAC helps make a smooth surrogate landscape (corresponding to Fig 2 right).
>
> **Why loss landscape is smoother?** The smoothness of the surrogate landscape is achieved by using the terminal value function after a short-horizon trajectory which absorbs the noisy landscape over long dynamics horizon and the discontinuity introduced by the early termination.
>
> **Histograms of the gradient distribution over time.** Thank you for this suggestion. We have added this analysis figure in the Appendix A.5 (Fig 12). Specifically, we pick six policies during the training on the *Humanoid* problem and plot the histograms of the gradient distribution for BPTT and SHAC respectively. The plots show that the gradient explosion problem exists in the BPTT method while the gradient acquired by SHAC is much more stable throughout the whole training process.
>
> &nbsp;
>
> **Q: How important is the structure of the reward function for this approach?**
>
> **A:** The reward function should be a differentiable function of the state and actions due to the use of a differentiable pipeline. As shown in Appendix A.3, We follow the reward function definitions from the latest RL benchmark work ([Isaac Gym](https://arxiv.org/pdf/2108.10470.pdf)) with the necessary modifications of converting non-differentiable terms into their differentiable approximation.
>
> &nbsp;
>
> **Q: The truncation of the curves in Fig 4 is visually misleading at first.**
>
> **A:** We understand the reviewer’s point. We have used truncation because our goal is to compare both the sample efficiency and wall clock time efficiency, but each algorithm may have its own advantage on either aspect. For example, PPO can use large amounts of samples but still be very fast in wall clock time by massively parallelizing the simulation over GPU as shown in [Isaac Gym](https://arxiv.org/pdf/2108.10470.pdf) work. On the other side, SAC can be more sample efficient than PPO but spending longer wall clock time due to its off-policy strategy. Given that, we have tried our best to balance the clarity of Fig 4 and its completeness by truncating the curves up to the maximal simulation steps/wall-clock time of our method. At the same time, we provide the full plot of the comparison in Appendix A.4.1 for reader’s reference.

---

### Author Response · Authors · 2021-11-15
**General Response to All Reviewers (1)**

We thank the reviewers for their insightful feedback. We are glad that the reviewers found that:

1. Our idea is simple (Reviewer [4PEt](https://openreview.net/forum?id=ZSKRQMvttc&noteId=tMN_P29pV-)), interesting (Reviewer [Hcaa](https://openreview.net/forum?id=ZSKRQMvttc&noteId=rMO1bWh85-)), and well-motivated (Reviewer [5Q2y](https://openreview.net/forum?id=ZSKRQMvttc&noteId=L72ig3zxfB)).
2. The GPU-parallized differentiable simulation with the associated benchmark set is a valuable contribution to the community. (Reviewer [Hcaa](https://openreview.net/forum?id=ZSKRQMvttc&noteId=rMO1bWh85-), [nSwn](https://openreview.net/forum?id=ZSKRQMvttc&noteId=3IzY97_Ggc))
3. The paper shows very impressive and convincing results. (Reviewer [4PEt](https://openreview.net/forum?id=ZSKRQMvttc&noteId=tMN_P29pV-), [Hcaa](https://openreview.net/forum?id=ZSKRQMvttc&noteId=rMO1bWh85-), [nSwn](https://openreview.net/forum?id=ZSKRQMvttc&noteId=3IzY97_Ggc), [5Q2y](https://openreview.net/forum?id=ZSKRQMvttc&noteId=L72ig3zxfB))
4. The experiments and ablation study are well designed. (Reviewer [5Q2y](https://openreview.net/forum?id=ZSKRQMvttc&noteId=L72ig3zxfB))
5. The manuscript is well-written and easy to follow. (Reviewer [Hcaa](https://openreview.net/forum?id=ZSKRQMvttc&noteId=rMO1bWh85-), [nSwn](https://openreview.net/forum?id=ZSKRQMvttc&noteId=3IzY97_Ggc), [5Q2y](https://openreview.net/forum?id=ZSKRQMvttc&noteId=L72ig3zxfB))

We have addressed the reviewers’ concerns individually. We also make the updates to the paper as follows (major changes are highlighted in red in the updated paper) (updated on Nov 20):

1. We have incorporated most writing and reference suggestions in the updated paper.
2. We have added more simulation details in Section 3.1 as suggested by Reviewer [Hcaa](https://openreview.net/forum?id=ZSKRQMvttc&noteId=rMO1bWh85-) and [nSwn](https://openreview.net/forum?id=ZSKRQMvttc&noteId=3IzY97_Ggc).
3. We have added two new environments, *HalfCheetah* and *Hopper,* in the benchmark and added the results in Figure 7 in Appendix A.4.1.
4. We have added the comparison with REDQ (a model-free RL method suggested by Reviewer [Hcaa](https://openreview.net/forum?id=ZSKRQMvttc&noteId=rMO1bWh85-)) and MBPO in Figure 10 in Appendix A.4.5 and Appendix A.4.6.
5. We have added the extra experiments suggested by the reviewers including experimenting with fixed policy architecture (Reviewer [nSwn](https://openreview.net/forum?id=ZSKRQMvttc&noteId=3IzY97_Ggc), Figure 8 in Appendix A.4.3) and experimenting with deterministic policy (Reviewer [5Q2y](https://openreview.net/forum?id=ZSKRQMvttc&noteId=L72ig3zxfB), Figure 9 in Appendix A.4.4).
6. We have added a plot of the comparison of finite difference single-weight gradients and analytical single-weight gradients in Figure 11 in Appendix A.5 to address Reviewer [5Q2y](https://openreview.net/forum?id=ZSKRQMvttc&noteId=L72ig3zxfB)’s concern.
7. We have added a plot of histogram of gradient distribution evolving over time as suggested by Reviewer [4PEt](https://openreview.net/forum?id=ZSKRQMvttc&noteId=tMN_P29pV-) in Figure 12 in Appendix A.5.
8. We have added a plot of landscape surface as suggested by Reviewer [nSwn](https://openreview.net/forum?id=ZSKRQMvttc&noteId=3IzY97_Ggc) in Figure 13 in Appendix A.5.
9. We have added a discussion about the comparison with model-based RL in related work section and in Appendix A.4.6.
---
To address some common concerns, in the following part of this general response, we

1. clarify the contribution of our work, and
2. compare our method to the model-based RL methods with similar ideas.

---

> ### Author Response · Authors · 2021-11-15
> **General Response to All Reviewers (2)**
>
> ### Contribution of our work
>
> Despite some similarities with previous works in model-based RL ([MAAC](https://openreview.net/pdf?id=Skln2A4YDB), [Dreamer](https://openreview.net/pdf?id=S1lOTC4tDS)), our paper and method (SHAC) is unique in the algorithmic and empirical insights it provides regarding how to combine an on-policy Actor-Critic method with a high-performance differentiable simulator. Our contributions are:
>
> 1. **SHAC is the first method that effectively combines reinforcement learning with analytical gradient computation from a differentiable simulator, and shows for the first time the potential of differentiable simulation to a wide set of classical RL benchmark control problems.**
>
>    While the recent development of differentiable simulation opens up new possibilities in physics-based animation and robotics applications, naive use of gradient based methods suffers from problems with local minima and gradient explosion. To show effectiveness of the differentiable simulators, previous works typically customize some problems with relatively simple dynamics or short horizons, and apply naive BPTT (e.g. [ChainQueen](https://arxiv.org/pdf/1810.01054.pdf), [Plasticinelab](https://arxiv.org/pdf/2104.03311.pdf)) or some proposed methods (e.g. [PODS](http://proceedings.mlr.press/v139/mora21a/mora21a.pdf)) to show superior performance over RL algorithms. However, it has not been demonstrated that differentiable simulators can work successfully on control problems at the same complexity level that RL algorithms can solve, which still remains as an open problem (also commented by [Brax](https://arxiv.org/pdf/2106.13281.pdf) section 5.1.4).  **Our paper is a timely attempt to fill this gap** by  first analyzing the reasons behind the optimization difficulties and then presenting an effective idea which combines model-based RL with analytical gradient computation from differentiable physics to **solve control problems at complexity levels that have never been achieved by differentiable simulation based methods before** (also credited by Reviewer [5Q2y](https://openreview.net/forum?id=ZSKRQMvttc&noteId=L72ig3zxfB) and [4PEt](https://openreview.net/forum?id=ZSKRQMvttc&noteId=tMN_P29pV-)), and shows the potential of differentiable simulation to improve not only the sample efficiency but more importantly the overall training time when compared to traditional RL algorithms.
>
> 2. **We evaluate our method with three classes of baseline for policy learning: model-free RL, model-based RL (with learned model), and previous differentiable simulation based methods.**
>
>    We compare our method to the following baseline methods:
>
>    - model-free RL: [a state-of-the-art implementation of PPO and SAC](https://github.com/Denys88/rl_games), and [REDQ](https://openreview.net/pdf?id=AY8zfZm0tDd) (as suggested by Reviewer [Hcaa](https://openreview.net/forum?id=ZSKRQMvttc&noteId=rMO1bWh85-))
>    - model-based RL: [MBPO](https://arxiv.org/pdf/1906.08253.pdf)
>    - differentiable simulation based methods: BPTT, [PODS](http://proceedings.mlr.press/v139/mora21a/mora21a.pdf), [SE-MBPO](http://proceedings.mlr.press/v139/qiao21a/qiao21a.pdf)
>
> 3. **We show that SHAC results in better empirical performance (either wall-clock time or sample efficiency or both, and higher final rewards) in five different environments of varying complexity.**
>
>    Our benchmark problems span a wide range of complexities. In the original manuscript, we include three classical RL benchmark problems (*CartPole Swing Up*, *Ant*, *Humanoid*), and a problem with 152-dimensional action space (*Humanoid MTU*). In addition, we add the *HalfCheetah* problem in the updated version as Reviewer [nSwn](https://openreview.net/forum?id=ZSKRQMvttc&noteId=3IzY97_Ggc) suggests, and will add *Hopper* problem in the later week.
>
> 4. **We will open source the GPU-parallelized differentiable simulator with the associated benchmark set, and also release the implementation of SHAC serving as a competitive baseline for the future research work in simulation-based policy learning.**

---

> > ### Author Response · Authors · 2021-11-15
> > **General Response to All Reviewers (3)**
> >
> > ### Compared to model-based RL methods
> >
> > We thank the reviewers for bringing additional related papers to our attention. Due to the simplicity & effectiveness of the actor-critic formulation, and under a broad definition that the simulator can be regarded as a blackbox “model’’, it is not surprising that the underlying idea of our method shares similarity with some prior model-based RL works ([MAAC](https://openreview.net/pdf?id=Skln2A4YDB), [Dreamer](https://openreview.net/pdf?id=S1lOTC4tDS)). While sharing some similarities, we believe the SHAC method proposed in this work has its own value and advantages over existing model-based RL algorithms for the following reasons:
> >
> > 1. **Our method (SHAC) is more wall-clock time efficient than model-based RL.**
> >
> >    MAAC and Dreamer need to learn the approximate dynamics model in order to provide a differentiable learned model. However, learning this approximate model itself is non-trivial and requires both computation and data (as shown in Section A.5 in [MAAC](https://openreview.net/pdf?id=Skln2A4YDB), it spends 24~37% of the total training time in learning the models for *HalfCheetah* and *Hopper*). In contrast, we leverage a differentiable simulator which obviates the need for model learning.
> >
> >    Furthermore, as the learned dynamics model in model-based RL is an approximation of the real simulation, model-based RL still requires many trajectory samples from this learned model and applies off-policy policy improvement updates, which also consumes computation (as shown in Section A.5 in [MAAC](https://openreview.net/pdf?id=Skln2A4YDB), it spends around 10 minutes on this step **per iteration**).
> >
> >    In contrast, due to the availability of the analytical derivatives from our differentiable simulator, our method can operate in on-policy mode requiring far fewer samples than the model rollouts required by model-based RL, resulting in a much higher wall-clock time efficiency. This intuition is supported by our new results for *HalfCheetah* where it only takes **less than 15 minutes** in total to successfully train a policy from scratch.
> >
> > 2. **Our method has improved policy performance than model-based RL methods.**
> >
> >    The learned dynamics model in model-based RL usually has model bias especially when the problem becomes complex, which prevents it from achieving the reward as high as training the policy directly in the simulator. One reason for this bias has been explored in prior work is [model mismatch](https://openreview.net/forum?id=4-D6CZkRXxI) or [objective mismatch](https://arxiv.org/abs/2002.04523) in model and policy learning steps.
> >
> > Based on those points, we think model-free RL or differentiable simulation based methods are more suitable choices where the policy learning happens in a cheap simulated environment, rather than in an expensive simulation or on the real robot.
> >
> >
> > **Comparison to MBPO**
> > To further illustrate the advantage of our method over model-based RL, we run a model-based RL method on our benchmark problems. Since MAAC (as mentioned by [Hcaa](https://openreview.net/forum?id=ZSKRQMvttc&noteId=rMO1bWh85-), [nSwn](https://openreview.net/forum?id=ZSKRQMvttc&noteId=3IzY97_Ggc)) does not provide an open source implementation, we are comparing SHAC with [MBPO](https://arxiv.org/pdf/1906.08253.pdf) (one of the standard model-based RL methods). We report results in Appendix A.4.6 from the [official MBPO](https://github.com/JannerM/mbpo), however we find its performance on our benchmark problems to be very poor (only works for the *CartPole Swing Up* problem).
> >
> > **Comparison to REDQ**
> > Furthermore, we have added comparisons between SHAC and [REDQ](https://openreview.net/pdf?id=AY8zfZm0tDd), a model-free RL method as requested by (Reviewer [Hcaa](https://openreview.net/forum?id=ZSKRQMvttc&noteId=rMO1bWh85-)). We find that REDQ is slow in wall-clock comparisons (e.g. in Ant problem, REDQ takes 27 hours to achieve 5000 rewards while SHAC only takes 10 minutes to achieve the same reward on the same machine). REDQ authors report that their method is up to 75% faster in wall-clock time from MBPO as it is (last paragraph section 2.1 in [REDQ](https://arxiv.org/pdf/2101.05982.pdf)). This experiment implicitly indicates the superior performance in wall clock time efficiency and final policy reward of SHAC over model-based RL.
> >
> > Furthermore, we also added the reference of MAAC and added a discussion about model-based RL in the related work section and in Appendix A.4.6.

---

> > > ### Author Response · Authors · 2021-11-17
> > > **Add Hopper and incorporate the new writing suggestions by Reviewer 5Q2y**
> > >
> > > 1. We added the *Hopper* environment in Appendix A.3.3 and add the experiment results in Figure 7 in Appendix A.4.1.
> > >
> > > 2. We also incorporate the new writing suggestions from Reviewer [5Q2y](https://openreview.net/forum?id=ZSKRQMvttc&noteId=NF65R-8sbaDD) to further clarify the algorithm and refer the reader in the main paper to the newly generated results in Appendix.

---

### Decision · Program_Chairs · 2022-01-20

**Decision:**

Accept (Poster)

**Comment:**

There is consensus in the reviews that this paper convincingly demonstrates strong acceleration of policy learning using differentiable simulation in tasks involving contact-rich dynamics. The authors are encouraged to explore where the smoothness assumptions made in the simulator actually transfer to real robots. The paper may be further strengthened through more complex benchmarks involving contact rich manipulation.